# DDM$^2$: Self-Supervised Diffusion MRI Denoising with Generative Diffusion Models

**Tiange Xiang, Mahmut Yurt, Ali B Syed, Kawin Setsompop & Akshay Chaudhari**
Stanford University
{xtiange, myurt, alibsyed, kawins, akshaysc}@stanford.edu

## Abstract

Magnetic resonance imaging (MRI) is a common and life-saving medical imaging technique. However, acquiring high signal-to-noise ratio MRI scans requires long scan times, resulting in increased costs and patient discomfort, and decreased throughput. Thus, there is great interest in denoising MRI scans, especially for the subtype of diffusion MRI scans that are severely SNR-limited. While most prior MRI denoising methods are supervised in nature, acquiring supervised training datasets for the multitude of anatomies, MRI scanners, and scan parameters proves impractical. Here, we propose **D**enoising **D**iffusion **M**odels for **D**enoising **D**iffusion **MRI** (DDM$^2$), a self-supervised denoising method for MRI denoising using diffusion denoising generative models. Our three-stage framework integrates statistic-based denoising theory into diffusion models and performs denoising through conditional generation. During inference, we represent input noisy measurements as a sample from an intermediate posterior distribution within the diffusion Markov chain. We conduct experiments on 4 real-world *in-vivo* diffusion MRI datasets and show that our DDM$^2$ demonstrates superior denoising performances ascertained with clinically-relevant visual qualitative and quantitative metrics. Our source codes are available at: https://github.com/StanfordMIMI/DDM2.

## 1 Introduction

Magnetic resonance imaging (MRI) is a non-invasive clinical imaging modality that can provide life-saving diagnostic information. Diffusion MRI is a subtype of MRI commonly used in oncologic and neurologic disorders (Bihan, 2003; Bihan et al., 2006), which can quantitatively assess micro-structural anatomical details. However, diffusion MRI scans suffer from severe signal to noise ratio (SNR) deficits, hindering diagnostic and quantitative accuracy. Image SNR can be improved either by lowering image resolution, which further reduces diagnostic utility, or by increasing the total scan time, which already can require 10+ minutes of time in the MRI scanner. Thus, there is large interest in decreasing diffusion MRI scan times to improve patient throughout in hospitals and the patient experience. While image SNR is governed by the underlying MRI physics, applying post-processing denoising techniques to fast and low-SNR MRI acquisitions can improve overall image SNR. Developing such methods to improve SNR of diffusion MRI scans is an unsolved problem that may improve the efficacy of the millions of such scans performed in routine clinical practice annually.

Supervised machine learning techniques have previously been proposed for MRI denoising, however, such methods are limited by their clinical feasibility. It is clinically impractical to acquire paired high- and low-SNR diffusion MRI scans across various anatomies (e.g., brain, abdomen, etc), diffusion weighting factors (and resultant SNR variations), MRI field strength and vendors, and clinical use-cases. Such large distributional shifts across heterogeneous use-cases leads to fundamental drop in model performance (Darestani et al., 2021). The diversity of data and the need for effective denoising methods motivates the use of unsupervised denoising techniques for diffusion MRI.

To address these challenges, the major contributions of our work are three-fold: *(i)* We propose DDM$^2$ for unsupervised denoising of diffusion MRI scans using diffusion denoising models. Our three-stage self-supervised approach couples statistical self-denoising techniques into the diffusion models (shown in Figure 1 Left); *(ii)* DDM$^2$ allows representing noisy inputs as samples from an intermediate state in the diffusion Markov chain to generate fine-grained denoised images without requiring ground truth references; *(iii)* We evaluate our method on four real-world diffusion MRI datasets

that encompass the latest and longer established MRI acquisition methods. DDM$^2$ demonstrates state-of-the-art denoising performance that outperforms second best runner-up by an average of 3.2 SNR and 3.1 contrast-to-noise ratio (CNR) points, across this large range of MRI distributions.

## 2 RELATED WORK AND BACKGROUND

### 2.1 DIFFUSION MRI

Diffusion MRI sensitizes the MR signal to the movement of protons within an object being imaged using tailored imaging gradients (small steerable magnetic fields). Given that both magnitude (overall duration that gradients are turned on, termed as *b-value*) and directions (linear combination of the three independent gradient directions) can be steered, the diffusion weighting is considered a vector. A typical diffusion MRI scan for resolving tissue microstructure is 4D in nature, with 3 dimensions of spatial coordinates and 1 dimension of diffusion vectors. Depending on the clinical use-case, the number of diffusion vectors can range from 2 to upwards of 100. Prior approaches for improving diffusion MRI sequences exist; however, these require highly-task-specific models (Kaye et al., 2020; Gibbons et al., 2018). These methods scan the same volume multiple times, average these multiple low-SNR images to generate a high-SNR image, and subsequently, train a supervised model to transform a single low-SNR image to the averaged high-SNR image. Such methods cannot generalizable to diverse anatomies, MRI scanners, and image parameters of diffusion MRI scans.

### 2.2 STATISTIC-BASED IMAGE DENOISING

Given a noisy measurement $\mathbf{x}$, an inverse problem is to recover the clean image $\mathbf{y}$ defined by:

$$\mathbf{x} = \lambda_1 \mathbf{y} + \epsilon, \tag{1}$$

where $\lambda_1$ is a linear coefficient and $\epsilon$ denotes the additive noise. Without losing generality, one usually represent $\epsilon = \lambda_2 \mathbf{z}$ as a sample from the Gaussian distribution, $\mathcal{N}(\mathbf{0}, \lambda_2^2 \mathbf{I})$. When paired ground truth images $\mathbf{y}$ are available, a neural network can then be trained to approximate $\mathbf{y}$ through direct supervisions. However, supervised training proves infeasible when $\mathbf{y}$ are missing in the datasets. Attempts have been made to relax the reliance on supervised signal from clean images, by using noisy images alone. Based on the assumption that the additive noise $\epsilon$ is pixel-wise independent, Noise2Noise (Lehtinen et al., 2018) claimed that to denoise an noisy measurement $\mathbf{x}$ towards another measurement $\mathbf{x}'$ is statistically equivalent to the supervised training on $\mathbf{y}$ up to a constant:

$$\mathcal{L}(\Phi(\mathbf{x}'), \mathbf{x}) = ||\Phi(\mathbf{x}') - \mathbf{x}||^2 \approx ||\Phi(\mathbf{x}') - \mathbf{y}||^2 + \texttt{const}. \tag{2}$$

Based on the above idea, Noise2Self (Batson & Royer, 2019) further designed the $\mathcal{J}$-Invariance theory that achieves self-supervised denoising by using the input $\mathbf{x}$ itself. With the statistical independence characteristic of noise, Noise2Void (Krull et al., 2019) and Laine *et al.* (Laine et al., 2019) extended the self-supervised strategy to a so-called 'blind-spot' technique that masked out image patches and designed a particular neural networks to predict the unmasked pixels. Neighbor2Neighbor (Huang et al., 2021) creates noisy image pairs by dividing sub-samples and uses them to supervise each other. Without losing pixel information, Recorrupted2Recorrupted (Pang et al., 2021) was proposed to keep all pixels intact and achieve self-supervised denoising by deriving two additional corrupted signals. To better utilize 4D MRI characteristics, Patch2Self (Fadnavis et al., 2020) was proposed to utilize multiple diffusion vector volumes as input $\{\mathbf{x}'\}$ and learn volume-wise denoising.

Building on the statistical theory studied above, in this work we achieve unsupervised MRI denoising through a generative approach. Compared to Patch2Self that requires a large number of volumes (e.g. $> 60$) as a guarantee to denoise a single volume, DDM$^2$ can efficiently denoise MRI scans acquired with very few diffusion directions and limited number of volumes (e.g. $< 5$). This is clinically relevant since common clinical diffusion MRI scans use fewer than 10 directions. Moreover, unlike Patch2Self, our trained denoiser is universally applicable for the entire 4D sequence and no repeated training is needed for denoising volumes in different directions.

### 2.3 DIFFUSION GENERATIVE MODEL

A diffusion model (Sohl-Dickstein et al., 2015) denotes a parameterized Markov chain with $T$ discretized states $\mathrm{S}_{1,\cdots,T}$ that can be trained to generate samples to fit a given data distribution. Transition of this chain is bi-directional and is controlled by a pre-defined noise schedule $\beta_{1,\cdots,T}$.

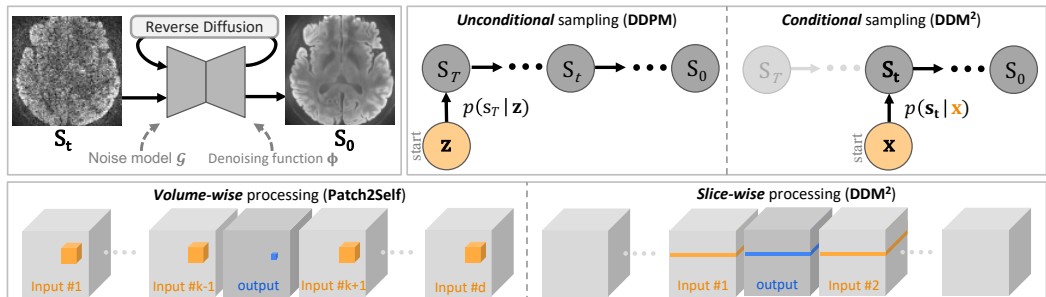

Figure 1: **Top Left:** DDM$^2$ generates clean MRI through reverse diffusion. **Top Right:** *Unconditional* sampling process (Ho et al., 2020) (starts from a noisy prior **z**) *v.s.* our *conditional* sampling process (starts from a noisy measurement **x**). **Bottom:** *Volume-wise* processing (Fadnavis et al., 2020) (requires a large number of input volumes) *v.s.* our *slice-wise* processing (requires very few input volumes). Gray blocks indicate accessible 3D volumes in a MRI sequence, orange blocks indicate the input priors $\{\mathbf{x}'\}$ and blue blocks indicate the one-pass output.

The forward process or the *diffusion process* $q(\mathrm{S}_t|\mathrm{S}_{t-1})$ gradually adds Gaussian noise to the posterior samples at every state until the signal is completely corrupted. The reverse of the above diffusion process or the *reverse process* tries to recover the prior signal with a parameterized neural network $\mathcal{F}$:

$$p_{\mathcal{F}}(\mathrm{S}_{0,\cdots,T}) \coloneqq p(\mathrm{S}_T)\prod_{t=1}^{T} p_{\mathcal{F}}(\mathrm{S}_{t-1}|\mathrm{S}_t), \qquad p_{\mathcal{F}}(\mathrm{S}_{t-1}|\mathrm{S}_t) \coloneqq \mathcal{N}(\mathrm{S}_{t-1};\mathcal{F}(\mathrm{S}_t,*),\sigma_t^2\mathbf{I}). \quad (3)$$

Starting from $\mathrm{S}_T = \mathbf{z} \sim \mathcal{N}(\mathbf{0},\mathbf{I})$, the reverse transition continues till the first state $\mathrm{S}_0$: a valid sample from the given data distribution. Following (Ho et al., 2020), we set $\sigma_t^2 = \beta_t$ and hold $\beta_{1,\cdots,T}$ as hyperparameters. In what follows, we refer this kind of set up as the baseline diffusion model.

Recent works have extended diffusion models to improve sampling quality or efficiency. Song *et al.* (Song & Ermon, 2019) investigated the density of data distribution and incorporated Langevin dynamics and score matching methods (Hyvärinen, 2005) into diffusion model. In a follow-up work, Song *et al.* (Song et al., 2021b) formulated score based generative models as solutions to Stochastic Differential Equation (SDE). Recently, Noise2Score (Kim & Ye, 2021) also trained a score function and in a manner similar to ours to perform score-based diffusion model for blind image denoising. Rather than transiting through the diffusion Markov chain, the authors validated its ability as a denoiser by following the Tweedie's formula (Efron, 2011). On the other hand, Denoising Diffusion Implicit Model (DDIM) (Song et al., 2021a) presented a non-Markovian acceleration technique for the reverse process that enables the sampling of a data in only few states. Similarly, Watson *et al.* (Watson et al., 2022) proposed a generalized family of probabilistic samplers called Generalized Gaussian Diffusion Processes (GGDP) to reduce the number of states without compromising generation quality. By using knowledge distillation, Salimans *et al.* proved that faster reverse process can be learnt from a slower one in the student-teacher manner iteratively (Salimans & Ho, 2022). Instead of starting from $\mathrm{S}_T$, our method initiates the reverse process from an intermediate state that also accelerates generation. DDM$^2$ is compatible with the above advances that can be utilized for further accelerations.

Prior works also adopted the diffusion model to recover data across modalities. WaveGrad (Chen et al., 2021), DiffWave (Kong et al., 2021), and BBDM (Lam et al., 2022) are for speech/audio generation. Luo *et al.* (Luo & Hu, 2021), Zhou *et al.* (Zhou et al., 2021), and GeoDiff (Xu et al., 2022) are for graph-structure data generation. Chung *et al.* (Chung et al., 2021) and Song *et al.* (Song et al., 2022) are for medical image inverse problem solving. Chung *et al.* (Chung et al., 2022) concurrently explored sampling noisy inputs as posteriors in the diffusion Markov chain for image denoising and super resolution using coarse noise approximations via image eigenvalues. In contrast, we integrate statistical theory and self-supervised noise estimation into diffusion models to account for noise distribution shifts. We note that there was no open source code available for (Chung et al., 2022) for using it as a comparison method.

## 3 Methods

### 3.1 Problem Definition and Overview

In this section, we show that clean image approximations $\bar{\mathbf{y}}$ can be generated through a diffusion model with a known noise model $\mathcal{G}$. To condition diffusion sampling on the noisy input $\mathbf{x}$, our idea is to represent $\mathbf{x}$ as a sample from a posterior at some data-driven intermediate state $\mathbf{S_t}$ in the Markov chain and initiate the sampling process directly from $\mathbf{x}$ through $p(\mathbf{S_t}|\mathbf{x})$. A comparison between the *unconditional* sampling process (Ho et al., 2020) that starts from $p(\mathbf{S}_T|\mathbf{z})$, $\mathbf{z} \sim \mathcal{N}(\mathbf{0}, \mathbf{I})$ and our *conditional* sampling process is outlined in Figure 1 Right.

There are now three questions need answers: *(i)* How to find a noise model $\mathcal{G}$ in an unsupervised manner? *(ii)* How to match $\mathbf{x}$ to an intermediate state in the Markov chain? and *(iii)* How to train the diffusion model as a whole when both noise model and the best matched states are available? To mitigate parameter tuning, DDM$^2$ is designed in three sequential stages as solutions w.r.t the above questions: **(Stage I)** Noise model learning; **(Stage II)** Markov chain state matching; and **(Stage III)** Diffusion model training. The overall framework is outlined in Figure 2. All experiments below are performed on 2D slices, however 3D volume results are shown in the appendix.

### 3.2 Stage I: Noise Model Learning

It is a common assumption that noisy signals can be obtained by corrupting the clean signals with noise sampled from an explicit noise model (Prakash et al., 2021). Without prior knowledge on the corruption process, it is difficult to find an effective mapping between $\mathbf{x}$ and $\mathbf{y}$. Following the $\mathcal{J}$-Invariance theory, one can approximate a solution by learning a denoising neural network from the noisy images themselves. To be more specific, with some prior signals $\{\mathbf{x}'\}$ as input, a denoising function $\Phi$ can be trained to generate a clean approximation $\bar{\mathbf{y}}$ of $\mathbf{x} \notin \{\mathbf{x}'\}$ such that $\mathbf{y} \approx \bar{\mathbf{y}} = \Phi(\{\mathbf{x}'\})$.

For a 4D MRI sequence $X \in \mathcal{R}^{w \times h \times d \times l}$ with $l$ 3D volumes (acquired at diverse gradient directions) of size $w \times h \times d$, Patch2Self learns $\Phi$ by aggregating $l-1$ volume patches as $\{\mathbf{x}'\}$ and regressing one clean voxel (3D pixel) $\bar{\mathbf{y}} \in \mathcal{R}^{1 \times 1 \times 1}$ for the remaining volume (Fadnavis et al., 2020). This training process repeats for all of the 3D volumes in $X$ to obtain $l$ volume-specific denoisers. There are three major drawbacks to such a denoising strategy: *(i)* the denoising function $\Phi$ needs to be trained for each volume individually, which inevitably demands long and repeated training process for MRI sequences with several volumes. *(ii)* the denoising quality heavily relies on the size of $\{\mathbf{x}'\}$—number of accessible volumes in the 4D sequence (see studies in section 4.3); *(iii)* when inspecting in 2D slices, the denoised results yield significant spatial inconsistencies due to its volume-wise regression.

Inspired by the performance of Patch2Self, but considering its challenges, we learn a slice-to-slice mapping function for the entire sequence instead of a patch-to-voxel mapping. This enables $\Phi$ to regress an entire 2D slice at a time by taking also slices to form $\{\mathbf{x}'\}$. Such a simple modification overcomes the above three drawbacks: *(i)* only a single $\Phi$ needs to be trained for the entire sequence; *(ii)* denoising quality remains stable when forming $\{\mathbf{x}'\}$ with very few volumes (see studies in section 4.3); *(iii)* better spatial consistency can be observed in the denoised slices. A comparison between our modification and original Patch2Self is briefly shown in Figure 1 Bottom. We retain this slice-wise processing scheme throughout all three stages in our framework.

### 3.3 Stage II: Markov Chain State Matching

When an optimal mapping function is learned by $\Phi$, the noise model $\mathcal{G}$ can be obtained by fitting the approximated residual noise $\bar{\epsilon}$ to a Gaussian $\mathcal{N}(\sigma^2 \mathbf{I})$ with zero mean and a variable standard deviation $\sigma$. Without any restrictions, $\bar{\epsilon}$ not necessarily has a mean value of zero and a direct fitting will lead to distribution mean shift. In this way, we propose to explicitly complement $\bar{\epsilon}$'s mean value $\boldsymbol{\mu}_{\bar{\epsilon}} = \frac{1}{||\bar{\epsilon}||} \sum \bar{\epsilon}$ to be zero and calibrate the denoising result $\bar{\mathbf{y}}$ accordingly that balances Equation 1:

$$\bar{\epsilon} := \bar{\epsilon} - \boldsymbol{\mu}_{\bar{\epsilon}}, \qquad \bar{\mathbf{y}} := \bar{\mathbf{y}} + \frac{\lambda_2 \boldsymbol{\mu}_{\bar{\epsilon}}}{\lambda_1}. \qquad (4)$$

The complemented $\bar{\epsilon}$ can be then used to fit the $\mathcal{G}$ and estimate $\sigma$. Recall that in the diffusion model, a noise schedule $\beta_{1,\cdots,T}$ is pre-defined that represents the noise level at every state in the Markov

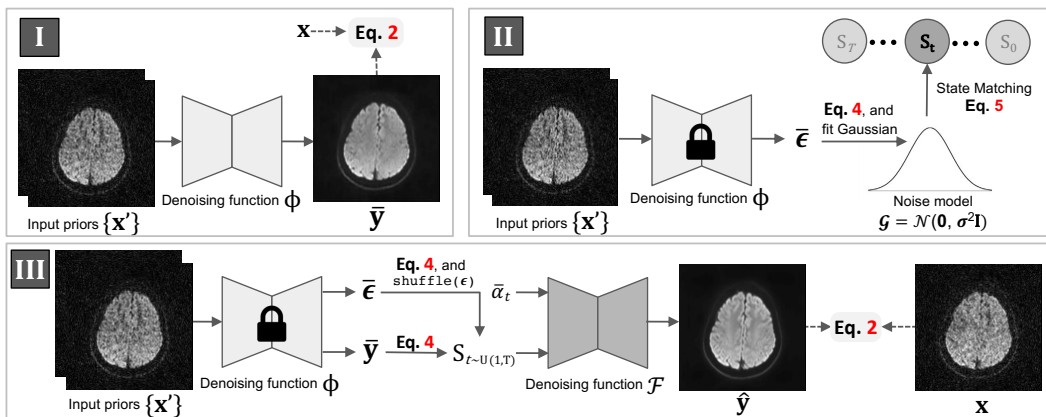

Figure 2: **Our three stage DDM$^2$.** In Stage I, we train a denoising function $\Phi$ in a self-supervised manner to estimate an initial noise distribution. In Stage II, the estimated noise by $\Phi$ is used to fit a Gaussian noise model $\mathcal{G}$. We match standard deviation of this noise model $\sigma$ to the diffusion sampling noise schedule $\beta$ and represent $\mathbf{x}$ as a sample at $S_{\mathbf{t}}$. With both $\mathcal{G}$ and $\mathbf{t}$, in Stage III, we train another denoising function $\mathcal{F}$ in the diffusion model for the reverse process $p_{\mathcal{F}}(S_0|\mathbf{x})$. All three stages take as inputs and returns 2D slices as discussed in section 3.2.

chain. We determine a matching state of $\mathbf{x}$ by comparing $\mathcal{G}$ with all possible posteriors $p(S_t)$—$\sigma$ and $\sqrt{\beta_t}$. Specifically, a state is matched when a time stamp $\mathbf{t}$ is found that minimizes the distance:

$$\arg\min_{t} ||\sqrt{\beta_t} - \sigma||^p, \tag{5}$$

where $|| \cdot ||^p$ denotes the $p$-norm distance. Since $t$ is a discrete integer within a finite interval: $\{1, \cdots, T\}$, we cast the optimization problem into a surrogate searching problem.

A match at state $S_{\mathbf{t}}$ indicates that, with this particular noise schedule $\beta$, there exists at least one possible sample from the posterior at state $S_{\mathbf{t}}$ in the baseline unconditional generation process that is sufficiently close to the given input $\mathbf{x}$. In this way, a denoised image can be sampled at state $S_0$ through the reverse process iteratively $p(S_0|S_t)$.

### 3.4 STAGE III: DIFFUSION MODEL TRAINING

In diffusion models, during the reverse process, an additional denoising function $\mathcal{F}$ is trained to predict $\bar{\epsilon}$ or $\bar{\mathbf{y}}$ at each transition step. As inputs to $\mathcal{F}$, clean images $\mathbf{y}$ are first corrupted through a prior sampling $q(S_t|\mathbf{y}) = \lambda_1\mathbf{y} + \lambda_2\mathbf{z}$, where $\mathbf{z} \sim \mathcal{N}(\mathbf{0}, \mathbf{I})$ (Equation 1). Following DDPM and its hyperparameters, we rewrite $\lambda_1 = \sqrt{\bar{\alpha}_t}$ and $\lambda_2 = \sqrt{1 - \bar{\alpha}_t}$ in terms of $\beta_t$: $\bar{\alpha}_t = \prod_{i=1}^{t} \alpha_i, \alpha_t = 1 - \beta_t$. In unsupervised denoising with $\mathbf{y}$ unavailable, denoised images $\bar{\mathbf{y}}$ are approximated by $\Phi$.

Unlike the supervised training schema that can train $\mathcal{F}$ on clean ground truth $\mathbf{y}$ directly, it is suboptimal to supervise $\mathcal{F}$ with the Stage I estimated $\bar{\mathbf{y}}$ in the unsupervised denoising scenario: $\mathcal{F}$ would degenerate to become an inverse function of $\Phi$ that traps $\mathcal{F}$ to interpolate only within the solution space of $\Phi$. To enable $\mathcal{F}$ extrapolation to a wider and more accurate solution space and to potentially improve denoising, we introduce two simple modifications as below.

**Noise shuffle.** First, we modify the sampling protocol for the diffusion process from $q(S_t|\bar{\mathbf{y}}) = \sqrt{\bar{\alpha}_t}\bar{\mathbf{y}} + \sqrt{1 - \bar{\alpha}_t}\mathbf{z}$ to $q(S_t|\bar{\mathbf{y}}) = \sqrt{\bar{\alpha}_t}\bar{\mathbf{y}} + \texttt{shuffle}(\bar{\epsilon})$, where $\texttt{shuffle}(\cdot)$ denotes the spatial shuffling operation and $\bar{\epsilon}$ is the residual noise predicted by $\Phi$. Under the assumption of noise independence, such noise shuffle operation forces $\mathcal{F}$ to learn from the implicit noise distribution modelled by $\Phi$ rather than the explicit noise model $\mathcal{G}$ to reduce the distribution disparity in between.

$\mathcal{J}$**-Invariance optimization.** Second, we modify the loss function to train $\mathcal{F}$. Borrowing the objective used in Stage I, we let $\mathcal{F}$ optimize towards the original noisy input $\mathbf{x}$ instead of $\bar{\mathbf{y}}$. The objective therefore becomes $\arg\min_{\mathcal{F}} ||\mathcal{F}(S_t, \bar{\alpha}_t) - \mathbf{x}||^2$. Such a modification not only minimizes the distance to the statistical 'ground truth' but also the approximation error brought by $\Phi$.

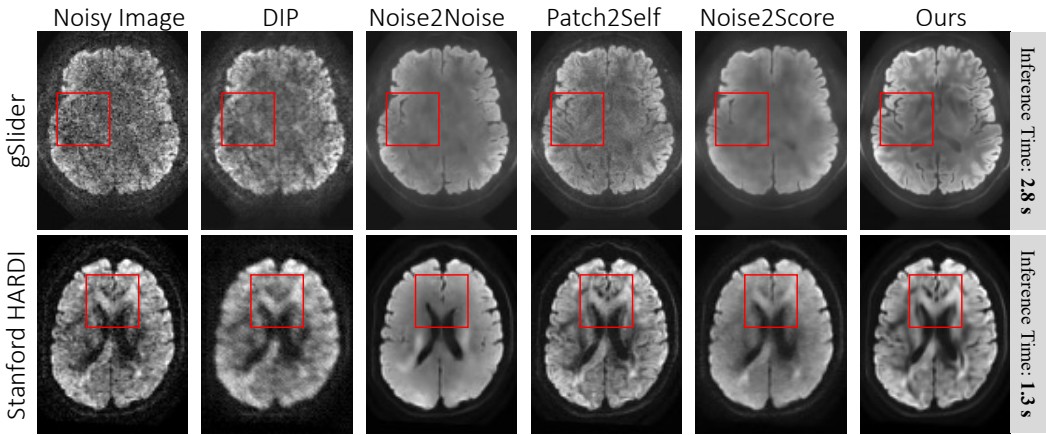

Figure 3: **Qualitative results on the most challenging datasets.** Inference time per slice is reported alongside. DDM$^2$ not only suppresses noise the most but also restores anatomical structures the best.

## 4 EXPERIMENTS

### 4.1 DATASETS AND EVALUATIONS

**Datasets.** Experiments were first conducted on an in-house dataset of brain diffusion MRI (gSlider) that has 0.5mm isotropic resolution, which was acquired under institutional review board approval using a recent advanced diffusion MRI sampling technique. The gSlider data had dimensions of $128 \times 128 \times 160 \times 50$ ($X \times Y \times Z \times$ *diffusion directions*) with b-value=1000. To evaluate the generalizability of DDM$^2$, additional experiments were done on 3 other publicly-available brain diffusion MRI datasets acquired with different protocols with less advanced MRI encoding for image SNR and resolution: *(i)* Sherbrooke 3-Shell dataset (dimensions = $128 \times 128 \times 64 \times 193$) with denoising occurring on the b-value=1000 images (amongst the available b-value=2000 and 3000 images) (Garyfallidis et al., 2014); *(ii)* Stanford HARDI (dimensions = $106 \times 81 \times 76 \times 150$) with b-value=2000 (Rokem, 2016); *(iii)* Parkinson's Progression Markers Initiative (PPMI) dataset (dimensions = $116 \times 116 \times 72 \times 64$) with b-value=2000(Marek et al., 2011). Experiments to add noise to raw k-space data for creating realistic noisy reconstructed images were performed on the SKM-TEA knee MRI dataset (Desai et al., 2021a) (*results in Appendix section G*).

**Evaluations.** We considered 4 state-of-the-art methods for comparisons with DDM$^2$: *(i)* Deep Image Prior (DIP), a classic self-supervised denoising method (Ulyanov et al., 2018); *(ii)* Noise2Noise (N2N), a statistic-based denoising method (Lehtinen et al., 2018); *(iii)* Patch2Self (P2S), state-of-the-art MRI denoising method; *(iv)* Noise2Score (N2S), a recent score-based denoising method. We note that assessing perceptual MRI quality is a challenging research problem (Mittal et al., 2011), as *there is a lack of consensus on image quality metrics, especially in an unsupervised reference-free settings* (Chaudhari et al., 2019) (Woodard & Carley-Spencer, 2006). Metrics such as PSNR and SSIM require a reference ground truth image, yet they still have poor concordance with perceptual image quality and consequently, evaluating on downstream clinical metrics of interest provides a better measure of real-world utility (Mason et al., 2020; Adamson et al., 2021). Thus, we utilized Signal-to-Noise Ratio (SNR) and Contrast-to-Noise Ratio (CNR) to quantify denoising performance and report mean and standard deviation scores for the full 4D volumes. As both metrics require regions of interest segmentation to delineate foreground and background signals, we follow the approach outlined in (Fadnavis et al., 2020). We segment the Corpus Callosum using (Garyfallidis et al., 2014) to generate quantitative and clinically-relevant Fractional-Anisotropy (FA) maps.

**Implementation details.** A typical noise schedule (Ho et al., 2020; Saharia et al., 2021) follows a 'warm-up' scheduling strategy. However, for most of MRI acquisitions, noise level is mild and the matched Markov chain state $S_t$ will be close to $S_0$. This leads to very few transitions in the reverse process and eventually poor generation quality. We therefore modify $\beta$ to follow a reverse 'warm-up' strategy such that $\beta$ remains at $5e^{-5}$ for the first 300 iterations and then linearly increases to $1e^{-2}$ between $(300, 1000]$ iterations. *See supplementary materials for more details.*

Figure 4: **Left:** Quantitative fractional anisotropy map comparisons (major differences highlighted with arrows). **Right:** DDM$^2$ demonstrates statistically significant SNR/CNR improvements (two-sided t-test, $< 1e^{-4}$ p-values) over all competing methods.

## 4.2 MAIN RESULTS

**Qualitative results.** In Figure 3, we visualize denoising results on axial 2D slices for each comparison method for the gSlider and HARDI datasets (additional dataset images in supplementary). Compared to DIP, all other methods demonstrate better MRI denoising abilities. Although the simplest Noise2Noise model and Noise2Score models achieve promising denoising, both suffer from excessive blurring of pertinent details in the underlying anatomical structures. Utilizing the information provided in all different volumes, Patch2Self is therefore able to restore most of the details. Unfortunately, this method is not robust to heavy noise and cannot produce clean estimations for the noisier Sherbrooke and gSlider datasets. Noise2Score, as a score-based denoising method, yields intermediate denoising quality between Noise2Noise and Patch2Self but with excessive blurring. By integrating statistical denoising into diffusion models, DDM$^2$ not only suppresses noise the most but also restores anatomical details. *Two neuroradiologists reviewed the DDM$^2$ images and commented that there were no new hallucinated lesions or regions with excessive blurring.*

**Quantitative results.** We calculated SNR/CNR on the entire Stanford HARDI dataset. The segmentation quality of the corpus callosum regions impacts metric calculation and an explicit tuning of RoIs would lead to unfair comparisons. To avoid this, we used scripts provided by (Garyfallidis et al., 2014) for RoI segmentation and metric calculation. In Figure 4 Right, we show box plots of relative SNR/CNR by subtracting the scores of the input noisy images from the denoised ones. Compared to DIP, N2N, and P2S, our method shows significant statistical improvements on both metrics (two-sided t-test, p-value $< 1^{-4}$). Compared to the most advanced score-based denoising method N2S, our method demonstrates an average of 0.95/0.93 improvements on SNR/CNR respectively.

The SNR/CNR measures depict how DDM$^2$ improves best-case results, depicting its excellent denoising capabilities, while maintaining worst-case results compared to the comparison methods. We note that N2S did have improved worst-case results than DDM$^2$ but this was likely due to excessive image blurring induced by the method (Figure 3). Note that without ground truth references, SNR is used as approximate indications of image quality.

As a diffusion model based approach, iterative inferences are necessary for DDM$^2$. This by its nature becomes a disadvantage from the perspective of inference time, especially when comparing to single forward pass-based CNN/MLP based methods. However, due to the novel design of Markov chain state matching, DDM$^2$ yields 5-10 times inference speed up than DDPMs. Moreover, since many diffusion MRI applications are performed in post-processing, improved SNR at the expense of a few minutes of inference can still benefit downstream clinical tasks.

**Fractional anisotropy comparisons.** Besides quantitative and qualitative evaluations with DDM$^2$ we examine how the denoising quality translates to downstream clinical tasks such as creation of Fractional Anisotropy (FA) maps using the various image denoising methods. To calculate FA, we used the volumes acquired by the first 6 diffusion directions along with the 10 b-value=0 volumes. Representative results displayed in Figure 4 Left indicate that the proposed DDM$^2$ yields enhanced and less noisy capture of diffusion directions of fiber tracts.

## 4.3 EXTENSIVE ABLATION STUDIES

**Modifications at `Stage I`.** In section 3.2, several upgrades were made upon Patch2Self for relaxing the reliance on large number of volumes (expensive to obtain during MRI acquisition) and for better

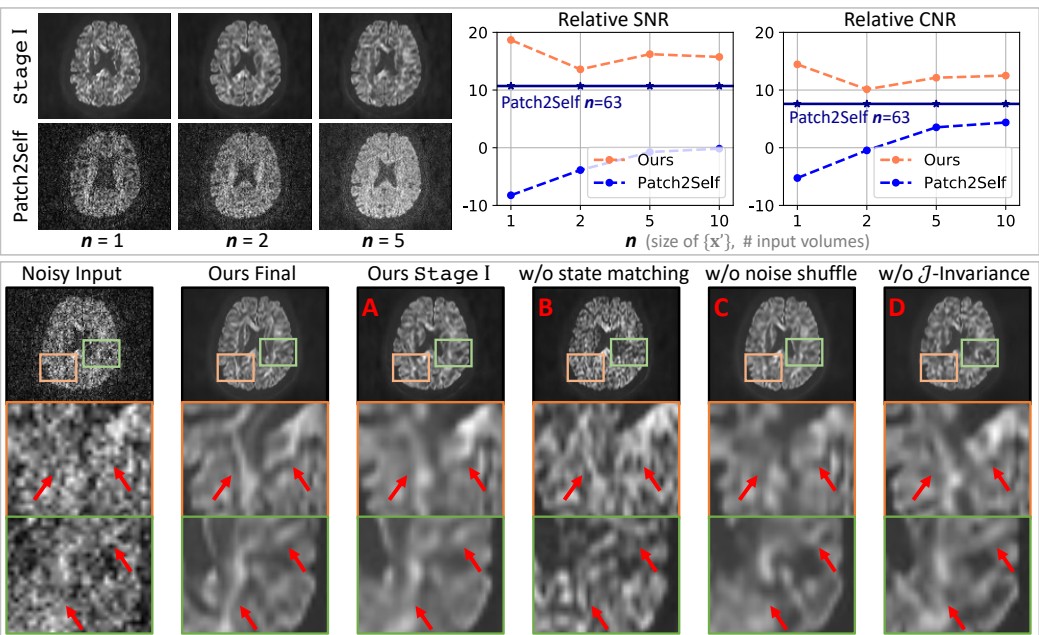

Figure 5: **Top:** Ablative results on number of accessible volumes $n$ used as inputs. **Bottom:** Qualitative results of the ablation studies. Results obtained by different ablation settings are labeled by different identifiers. Major differences are zoomed in and are highlighted by red arrows.

spatial consistency in the denoised results. In Figure 5 Top, we compare our `Stage I` against Patch2Self by inputting limited number of volumes $n$ (size of $\{x'\}$). Patch2Self can achieve decent denoising performance when using a large number of volumes (e.g. n>60). When $n$ is small, Patch2Self fails to denoise the data properly. However, our simple modifications leverage spatial priors and achieve impressive denoising result even for $n = 1$. With more accessible volumes, our `Stage I` maintains its denoising performance at a high level. Considering both denoising quality and clinical applicability, we adopted $n = 2$ as our final setting.

**Necessity of the multi-stage design.** With the learned noise model from `Stage I`, our method learns an additional diffusion model to restore more subtle details and reduce the approximated statistical error. To compare whether our three-stage method performs better than the `Stage I` alone, we compare our final results to the denoised results obtained at `Stage I` (Figure 5 **A**, *more comparisons can be found in supplementary materials*). Due to the upgrades made in section 3.2, our `Stage I` already achieves satisfactory denoising results. However, similar to other statistic-based methods, `Stage I` cannot restore high-frequency details and always converges to smooth solutions. This emphasized the novelty of using a diffusion model to generate self-consistent details to further restore the underlying anatomy.

**Necessity of `Stage II`.** In section 3.3, we propose to represent $\mathbf{x}$ as a sample from an intermediate state $S_t$ in the Markov chain. This is done by minimizing the distance between the fitted standard deviation and the pre-defined noise schedule $\beta$. Instead of finding the most matched state, we evaluate whether it is possible to naively represent $\mathbf{x}$ as the middle state $S_{t=\frac{T}{2}}$ regardless of $\beta$. Given that the reverse process starts from $\mathbf{z} \sim \mathcal{N}(\mathbf{0}, \mathbf{I})$, representing $\mathbf{x}$ as $S_{t=\frac{T}{2}}$ may overestimate the noise level. Consequently, excessive signals will be generated and be imposed into the denoising results as shown in Figure 5 **B**. These hallucinations are not valid restorations of the underlying anatomical structure even though they may lead to high SNR/CNR scores.

**Modifications at `Stage III`.** As mentioned in section 3.4, two modifications are designed to assist diffusion model training. We visualize the denoising results by disabling the noise shuffle operation (Figure 5 **C**) and the $\mathcal{J}$-Invariance optimization (Figure 5 **D**). By learning from an explicit noise model and optimizing towards the `Stage I` results, we observe significant structural degradation in the denoised images. Moreover, both of the results differ considerably from the `Stage I` results (Figure 5 **A**), which further validates the need of our modifications to be used in `Stage III`.

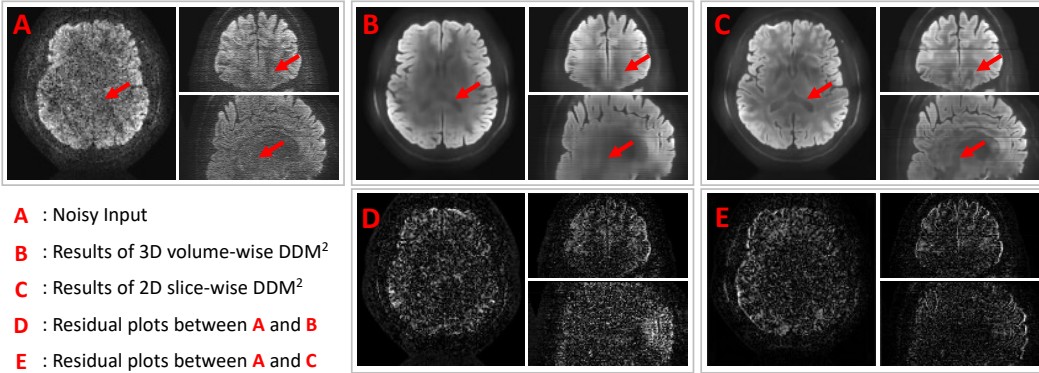

**A** : Noisy Input

**B** : Results of 3D volume-wise DDM²

**C** : Results of 2D slice-wise DDM²

**D** : Residual plots between **A** and **B**

**E** : Residual plots between **A** and **C**

Figure 6: Comparisons between 3D volume-wise DDM² and 2D slice-wise DDM². Results are shown on the transverse, Coronal, and Sagittal planes. Major differences are highlighted by arrows.

**3D volume-wise DDM².** Other than 2D slice-wise processing schema of DDM², 3D volume-wise schema could be another natural design. However, we show its inferior denoising performances via experiments as demonstrated in Figure 6. The residuals between the original and denoised images maintain more structure in the 3D methods than 2D methods, which depicts subpar denoising performance. To match computational burden of 2D slice-wise experiments, we experimented with the largest possible patch size of $16 \times 16 \times 16$. *In supplementary materials, we discussed the impact of patch size and presented additional results.* There are three drawbacks can be observed: *(i)* Volume-wise DDM² requires larger networks with $3\times$ model parameters; *(ii)* Contextual dependencies in the longitudinal dimension (across slices) might be weaker for datasets acquired with thick slices due to varying noise characteristics within and across slices. This limits 3D networks, whereas 2D networks can capture the information within the isotropic slice; *(iii)* Most advances in diffusion models focus on 2D sequential acquisitions, which can be easily adopted to improve DDM².

## 5 DISCUSSIONS

**MRI noise considerations.** Noise in MRI typically follows a Rician distribution for background regions and transitions to Gaussian in foreground regions where the SNR≥3 (Andersen, 1996; Robson et al., 2008). Moreover, linear image reconstructions result in complex Gaussian noise remaining Gaussian in the image domain also (Robson et al., 2008). Modern image acceleration techniques simply perturb the magnitude of noise spatially (based on coil and subject patterns) but to not modify the Gaussian distributions (Pruessmann et al., 1999). Consequently, using DDM² for denoising relevant foreground image regions is consistent with the underlying image noise statistics. For future work in non-linear image reconstructions (such as learning-based reconstructions (Chaudhari et al., 2020)), our framework still allows data-driven fitting of different noise profiles with a corresponding noise model and score function suggested by Noise2Score.

**Limitations.** As with most DDPM applications, DDM² is primarily limited by slow inference durations. However, one can switch to faster networks or explore acceleration techniques as in (Song et al., 2021a). Moreover, DDM² may hallucinate actual lesions due to its generation nature. While this was not seen in our results, it can be mitigated by enforcing data consistency during inference. Systematic radiologist-driven studies may also assess anatomic integrity and expert quality metrics.

## 6 CONCLUSION

In this paper, we present DDM², a self-supervised denoising method for MRI. Our three-stage method achieves MRI denoising via a generative approach that not only reduces noisy signals but also restores anatomical details. In `Stage I`, a denoising function is trained to capture an initial noise distribution. In `Stage II`, we fit a noise model to match the noisy input with an intermediate state in the diffusion Markov chain. In `Stage III`, we train a diffusion model to generate clean image approximations in an unsupervised manner. In experiments, we demonstrate that DDM² outperforms existing state-of-the-art denoisers for diffusion MRI methods both qualitatively and quantitatively.

ACKNOWLEDGEMENT

We thank Congyu Liao for his feedback and provision of the gSlider data. This work was supported by NIH grants R01 AR077604, R01 EB002524, and R01 AR079431. Additional research support was provided by the Stanford Radiological Sciences Laboratory Neuroimaging Research grant.

ETHICS STATEMENT

All datasets used in this study were acquired under institutional review board (IRB) approval for human subjects research. No additional metadata is provided to enable re identification of the individuals involved. The proposed denoising technique has the ability of reducing MRI scan time or enabling noisier data acquisitions on older MRI scanners. This presents a large opportunity to make life-saving MRI technology available and more accessible to patient populations that would not ordinarily be exposed to it due to initial infrastructure and maintenance costs of expensive MRI scanners. Further clinical validation of the proposed technique will be necessary prior to routine usage in these scenarios, however, in order to ensure lack of visual hallucinations and comparable diagnostic outcomes compared to the current standards of care.

REPRODUCIBILITY STATEMENT

Our proposed method is fully reproducible with respect to the main results as reported in section 4.2. For methodology & implementation details, readers are referred to our source codes, which are available at: `https://github.com/StanfordMIMI/DDM2`.

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

# A    TABLE OF SYMBOLS

For notation simplicity, we adopted alphabetic symbols in this paper to represent essential components in our framework. For better symbol-name correspondences, here we justify implications of all symbols used in the paper in Table 1 to help readers comprehend.

| Symbols | Implication |
|---------|-------------|
| $\mathbf{x}$ | target noisy input |
| $\mathbf{x}'$ | noisy input prior |
| $\mathbf{y}$ | clean ground truth of $\mathbf{x}$ |
| $\epsilon$ | residual noise |
| $\lambda$ | scale coefficients balancing $\mathbf{y}$ and Gaussian noise as in Equation 1 |
| $\bar{\mathbf{y}}$ | Stage I estimated clean image of $\mathbf{x}$ |
| $\bar{\epsilon}$ | Stage I estimated residual noise |
| $\mu_{\bar{\epsilon}}$ | arithmetic mean of $\bar{\epsilon}$ |
| $\phi(\cdot)$ | denoising function in Stage I |
| $\mathcal{F}(\cdot)$ | denoising function in Stage III |
| S | state in the Markov chain |
| $\mathcal{G}$ | noise model |
| $\sigma$ | standard deviation of noise distribution |
| $\beta$ | noise schedule in diffusion models |

Table 1: Table of symbols used in this paper.

# B    EXPERIMENTAL DETAILS

**Implementation details.** We implement both denoising functions $\Phi$ and $\mathcal{F}$ via U-Net (Ronneberger et al., 2015) with modifications suggested in (Song et al., 2021b; Saharia et al., 2021). In `Stage I` we set the size of $\{\mathbf{x}'\}$ $n = 2$. In `Stage III`, inspired by (Song & Ermon, 2019; Chen et al., 2021), we train $\mathcal{F}$ to condition on $\bar{\alpha}_t, t \sim U(1, T)$. The Adam optimizer was used to optimize both networks with a fixed learning rate of $1e^{-4}$ and a batch size of 32. We trained $\Phi$ for $1e^4$ steps and $\mathcal{F}$ for $1e^5$ steps from scratch. All experiments were performed on RTX GeForce 2080-Ti GPUs in PyTorch (Paszke et al., 2019).

**Evaluation details.** Our DDM$^2$ is compared against 4 state-of-the-art unsupervised denoising methods. *For fair comparisons, we aligned the total number of parameters in DDM$^2$ and all comparison methods: For U-Net based methods, we adopted the exact architecture as in DDM$^2$; For MLP based methods, we aligned the number of parameters. (i)* Deep Image Prior (Ulyanov et al., 2018) trains a neural network on random inputs to overfit a specific noisy image. The network is claimed to restore low-level consistencies exist in the target image at an intermediate state of the training process. In our experiments, since there is no reliable quantitative metrics like PSNR, we pause network training based on SNR and the visual quality of the reconstructions; *(ii)* Noise2Noise (Lehtinen et al., 2018) constructs training pairs of two independent noisy measurements of the same target and trains a network to transform one measurement to the other. 4D MRI by their nature own independent noisy samples acquired at different gradient directions. In our experiments, we train Noise2Noise model by using the same slices at different volumes; *(iii)* Patch2Self (Fadnavis et al., 2020) generalizes Noise2Noise and Noise2Self for patch-wise MRI denoising. In our experiments, we follow their official implementation and adopted MLP as the regressor and patch radiance 0 for their best performance; *(iv)* Noise2Score (Kim & Ye, 2021) trains a neural network to estimate a noise-dependent score function to find the clean posterior distribution of noise inputs. In our experiments, we estimate the score function under the same assumption of Gaussian noise.

# C    INTUITIVE DEMONSTRATION OF MARKOV CHAIN STATE MATCHING

In section 3.3, we claim that a matched state at $\mathbf{t}$ denotes the minimum $p$-norm distance achieved between $\sqrt{\beta_t}$ and $\sigma$ (the standard deviation of the Gaussian fitted on the input image $\mathbf{x}$). To verify the match intuitively, we show visual comparisons between $\mathbf{x}$ and the noise corrupted `stage I`

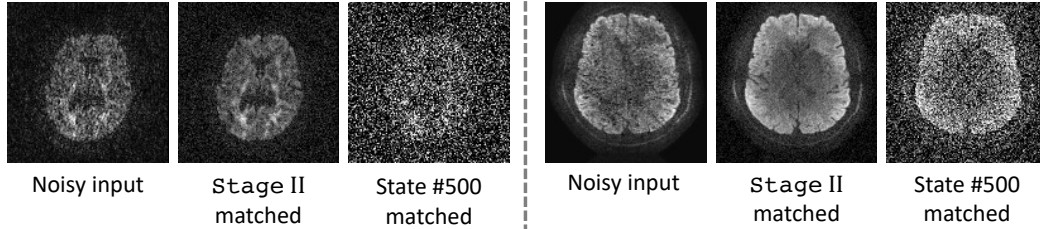

Figure 7: Visual demonstration of different matches in the Markov chain. Our `Stage II` matched state generates samples that look sufficiently close to the noisy input, while arbitrarily chosen states (e.g. state #500) cannot match the noisy input well.

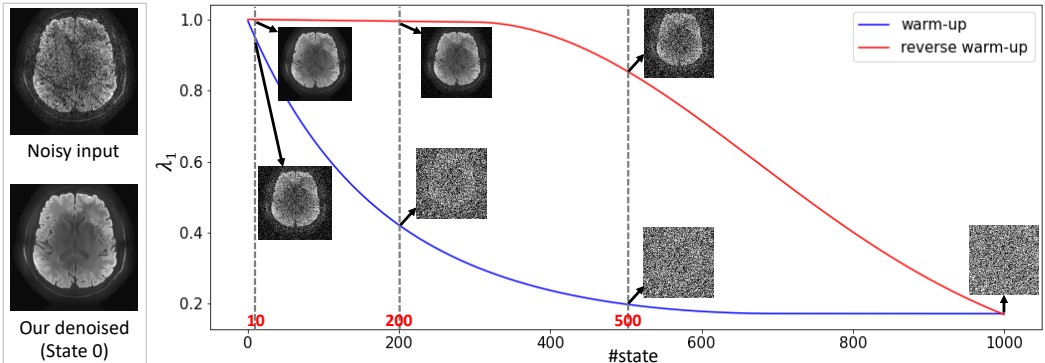

Figure 8: Comparison of the baseline warm-up based schedule and our reverse warm-up based schedule. Our modified schedule allows noisy inputs to be represented at a later state than the baseline, which leads to more transition steps and better denoising quality.

result $\mathbf{c} = q(\mathbf{S_t}|\mathbf{S_0}) = \sqrt{\bar{\alpha_t}}\bar{\mathbf{y}} + \epsilon$, $\epsilon \sim \mathcal{N}(\mathbf{0}, \beta_t\mathbf{I})$ as a prior sample at the state $\mathbf{S_t}$. A successful match leads to close image distributions ($\mathbf{x}$ and $\mathbf{c}$ in similar appearances), while $\mathbf{x}$ and $\mathbf{c}$ may look considerably different in unsuccessful matches.

As shown in Figure 7, compared to a naive representation at the middle Markov chain (state #500, as studied in section 4.3), our `stage II` matches the input in the Markov chain better. The successful match indicates there exists at least one possible sample drawn from the posterior distribution at the matched state that is sufficiently close to the noisy input. Therefore our claim in section 3.1 can be well proved. Without a proper state matching strategy, the final generation results will be in poor quality (section 4.3 **B**).

## D  NOISE SCHEDULE WITH REVERSE WARM-UP

Noise levels in most of MRI datasets are small. Matching them in `Stage II` with commonly used 'warm-up' based noise schedule $\beta$ leads to few transitions in the sampling process and eventually poor generation quality. Considering this, we adopted a 'reverse warm-up' setting for $\beta$.

We compare $\lambda_1^{(t)} = \sqrt{\prod_{i=1}^{t}(1 - \beta_t)}$ of the warm-up schedule ($\beta_1 = 1e^{-2}$, $\beta_T = 5e^{-5}$, ratio= 0.7) and our reverse warm-up schedule ($\beta_1 = 5e^{-5}$, $\beta_T = 1e^{-2}$, ratio= 0.7) in Figure 8. Recall that $\mathbf{x} = \lambda_1\mathbf{y} + \lambda_2\mathbf{z}$ (Eq. 1). During the sampling process, our reverse warm-up based strategy raises the value of $\lambda_1$ faster than the baseline strategy, hence reveals the underlying anatomical structures in MRIs much faster. This allows us to represent noisy inputs at a later state (e.g. state #200) in the Markov chain to enable more reverse process transitions. The baseline warm-up based schedule, on the other hand, represents noisy inputs at a much earlier state (e.g. state #5). In this way, the final denoising result will be generated in very few transitions and eventually in inferior quality.

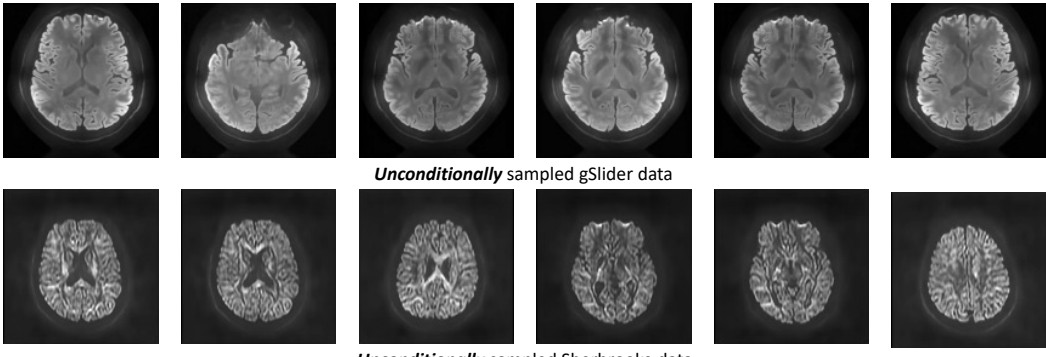

*Unconditionally* sampled gSlider data

*Unconditionally* sampled Sherbrooke data

Figure 9: Unconditional sampling of DDM$^2$ on our gSlider and the Sherbrooke datasets. All of the generations are realistic due to the restored anatomical details.

## E    UNCONDITIONAL SAMPLING OF DDM$^2$

Differing from all competing methods, besides functioning as a denoiser, our diffusion model based method is capable of generating new (nonexistent) images that fit a given dataset. We here conducted experiments to generate realistic MRI scans through *unconditional sampling* from our trained diffusion model. The generation results for our gSlider dataset and the Sherbrooke dataset are shown in Figure 9.

These forged images successfully restored essential anatomical details in the MRI scans, which can be used to enlarge the dataset further to assist down-streaming tasks.

## F    OUTLIER DETECTION

As a diffusion model based method, the denoising performance of DDM$^2$ is relatively input-dependent. In order to detect on which cases DDM$^2$ performs the best and which the worst, we here present a simple technique that can be adopted during DDM$^2$ inference to detect poorly denoised outliers.

Specifically, we compute RMSE scores between the initial noisy input ($S_t$) and every image ($S_{t \to 0}$) generated in the reverse diffusion process. For good denoising, RMSE values will increase significantly over iterations (represents denoised results dissimilar to inputs). For poorly denoised outliers, RMSE values will change slowly (represents denoising results similar to the original noisy measurement).

We provide demo results in Figure 10. Clearly, the poorly denoised slice (bottom slice) cannot achieve as high RMSE value as the well denoised one (top slice). This technique is easy in implementation and can be adopted to detect poorly denoised outliers by DDM$^2$ effectively.

## G    EVALUATION ON SIMULATED DATA

Apart from the experiments conducted on real-world datasets, we investigate whether DDM$^2$ can still recover clean anatomical patterns by simulating noisy k-space data. We leverage the SKM-TEA dataset Desai et al. (2021c) for implementing these simulations since it provides raw, complex, multi-echo, and multi-coil k-space. To simulate the impact of adding additional complex noise to the k-space data, we leverage k-space noise addition strategies that have been validated in prior work (Desai et al., 2021b; 2022). Briefly, we sample a random complex Gaussian noise with varying standard deviations (to simulate different SNR levels) and apply these noise maps to each coil k-space data. Subsequently, we perform image reconstruction to transform the k-space data into magnitude images with varying levels of noise. We simulated noisy images at 4 different SNR levels in total. We also experiment on how the number of prior volumes (i.e. n) impact the performance of Patch2Self and DDM$^2$ under these simulation settings. In this case, we conduct experiments with (n=1) and (n=10) prior volume. While no changes are made to the hyper-parameters of DDM$^2$, we follow

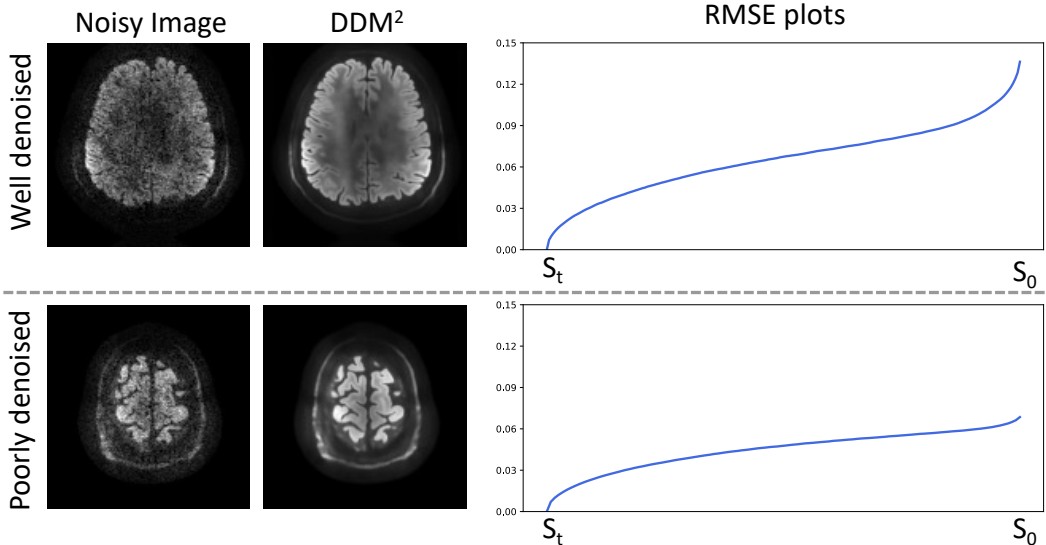

Figure 10: RMSE plot comparisons between well denoised and poorly denoised slices.

the suggestions in Patch2Self and increase its patch radius for both experiments. Since there is an available ground-truth to compare to, we compute quantitative (PSNR and SSIM metrics) results are reported for the simulated noisy images and the resulting DDM$^2$ and Patch2Self denoised volumes.

In Figure 11, we show the denoising results on the simulated noisy images with only n= 1 prior volume available. When input images are in a high SNR, DDM$^2$ demonstrates on-par results to Patch2Self. On low SNR images, DDM$^2$ surpasses Patch2Self significantly in terms of both visual quality and quantitative metrics. The same phenomenon can be also observed in Figure 12, where results are obtained with n= 10 prior volumes. Noteworthy, when the input images are highly corrupted (the left most column), DDM$^2$ still recovers a valid anatomical pattern due to its generative nature. Patch2Self, on the other hand, fails to denoise on low SNR inputs.

## H  ADDITIONAL COMPARISONS

**Compare with different Patch2Self settings.**  In the main paper, we re-implemented Patch2Self with MLP denoisors at the same network scale as DDM$^2$ for fair comparison. Here we compare against official Patch2Self implementations under different settings. Specifically, we compare with three different denoisors (MLP, OLS, Ridge) and explore on larger patch radius. Note that our CPUs crashed when patch radius is larger than 1, yielding its unaffordable computational costs.

Denoising results are shown in Figure 13. No significant differences can be observed from the results for Patch2Self with different denoisors. Although setting a larger patch radius improves the results slightly, computational costs increase cubicly and common CPUs with 32 GB memory cannot afford.

**Compare with classic Gaussian noise denoisors.**  Under our Gaussian noise assumption, one may wonder if classic denoising methods can be directly applied on dMRI data and our three-stage DDM$^2$ is not necessary. Here, we compare with two commonly used classic methods to prove the essence of DDM$^2$—Stein's Unbiased Risk Estimator (SURE) Metzler et al. (2018) and the Implicit Prior (IP) Kadkhodaie & Simoncelli (2020). Moreover, we also combine these two methods and use one to refine the results from another to see if performance boosts can be observed.

Denoising results are shown in Figure 14. As common denoisors for Gaussian noise, both SURE and IP fail to denoise dMRI signals and the combination of these two doesn't increase denoising quality

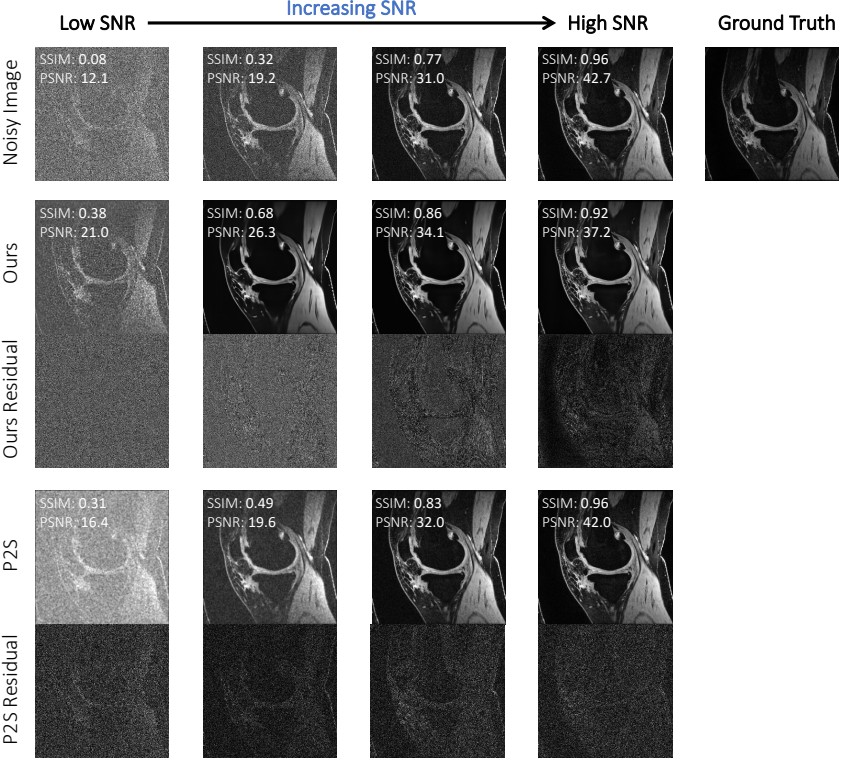

Figure 11: Results on simulated data with n= 1 shows that in low-noise (high SNR) settings, $DDM^2$ and Patch2Self exhibit comparable performance. However, as noise levels increase (lower SNR), $DDM^2$ maintains denoising performance, while Patch2Self is unable to perform denoising. All images best viewed in zoomed in.

as expected. We note that Noise2Score Kim & Ye (2021) is a recent method that extends SURE for better score estimation. After applying IP as a post-processor for Noise2Score, the final denoising results are still significantly inferior than ours. Therefore, we claim that, even with the Gaussian noise assumption, $DDM^2$ still learns meaningful noise distributions from the dataset itself, which can not be achieved by any of the comparison methods.

**Compare under mixed b-value images.** In the main paper, we mainly focused on homogeneous b-values since these represent the most clinical and research MRI protocols (where one b-value shell is sampled across multiple directions). However, we are also interested if denoising methods can function on images of mixed b-values. We adopt the Sherbrooke 3-Shell dataset for this experiment. In details, we train on a mixture of b-value=1000 and b-value=2000 images and try to denoise the b-value=2000 ones.

Denoising results in Figure 15 depict that the resultant images, although denoised, have a high amount of blurring. This is most likely due to the fact that different b-values have different gradient strengths, and as a result, different distortion artifacts induced by the echo planar readout. This distortion across b-values manifests as blurring in the denoised image.

## I ADDITIONAL RESULTS

**DTI Diffusivity results.** In addition to the factional anisotropy maps we presented in section 4.2 and Figure 4, here we perform additional DTI measurements of Axial Diffusivity (AD), Mean Diffusivity (MD), and Radial Diffusivity (RD) to further validate our $DDM^2$. According to the comparison

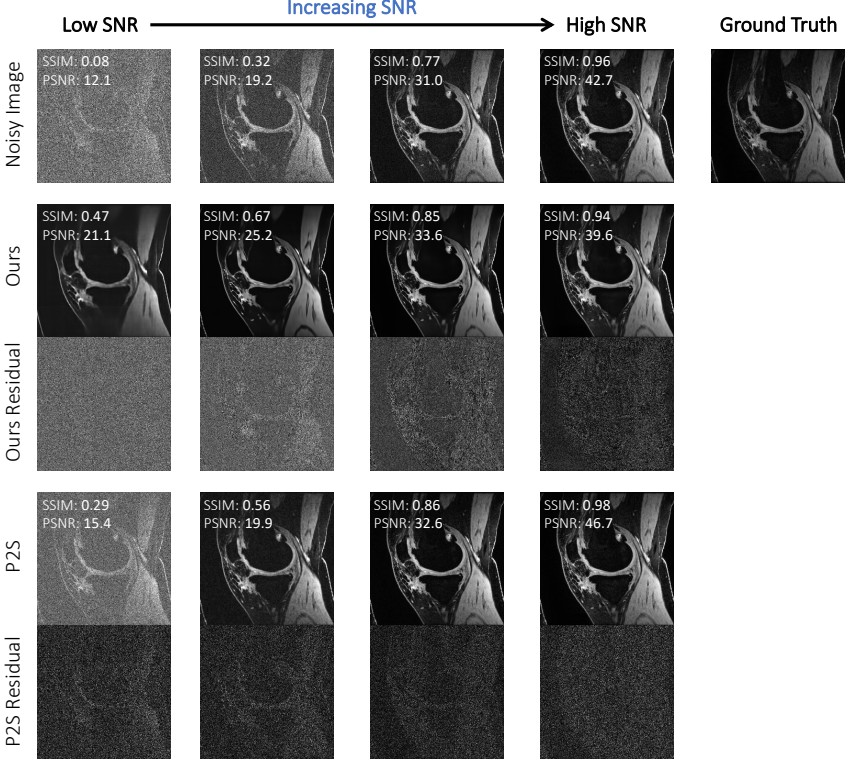

Figure 12: Results on simulated data with n= 10 supporting volumes. These images indicate that the performance of DDM$^2$ is comparable independent of how many supporting volumes are available, unlike Patch2Self which requires an increasing number of volumes that may not always be clinically available, to achieve reasonable performance. All images best viewed in zoomed in.

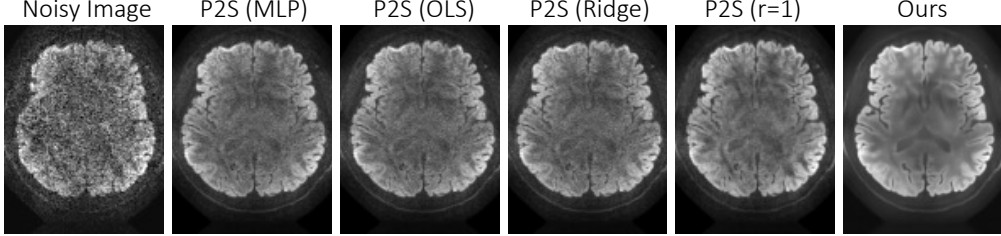

Figure 13: Additional comparison with different Patch2Self settings depict similar performance across the hyperparameter space for Patch2Self. All Patch2Self models still suffer from how SNR, compared to DDM$^2$.

results shown in Figure 16, both AD of Patch2Self and Noise2Score show very smooth signals outside the ventricles. However, DDM$^2$ shows a lot more of the natural structure.

**Qualitative results.** Additional qualitative comparisons on all of the four datasets are presented in Figure 17. Obviously, DDM$^2$ not only reduces noise the most but also restores anatomical details the best.

**3D volume-wise DDM$^2$.** In section 3.2, we present our slice-wise improvements over the patch-wise processing (Fadnavis et al., 2020). To validate our design choice, in section 5, we presented an alternative design of DDM$^2$ that functions on 3D volume patches rather than 2D slices. An intuitive comparison between the two designs is outlined in Figure 18.

The best results for 3D volume-wise DDM$^2$ were achieved at patch size $16 \times 16 \times 16$—also the largest volume patches we can afford in our GPU memory. Empirically, we found that when functioning on small-size patches, diffusion models fail to denoise but instead overfit and return the exact inputs.

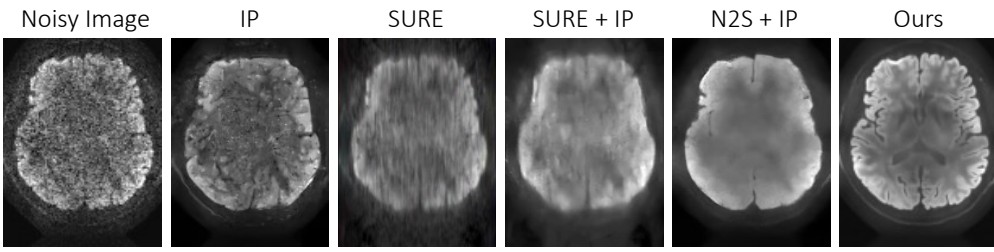

Figure 14: Additional comparison with classic Gaussian noise denoisors.

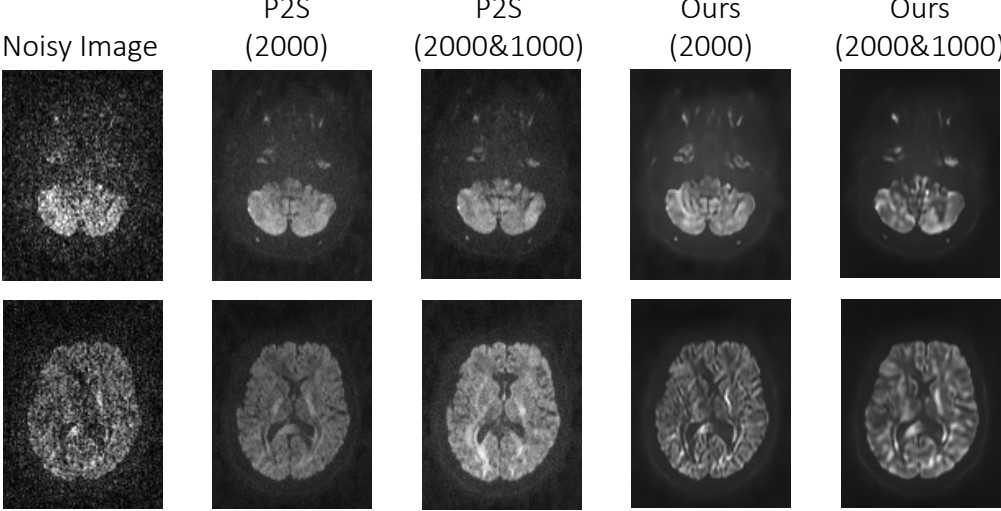

Figure 15: Additional results when training on mixed b-value images exhibit adequate denoising but excessive blurring, likely due to different distortion profiles between the b=1000 and b=2000 scans that lead to anatomical inconsistencies.

Comparisons at the transverse, coronal, and sagittal planes between the 2D slice-wise DDM$^2$ and 3D volume-wise DDM$^2$ are shown in Figure 6 and Figure 19.

**Stage I Comparisons.** In Figure 20 we present additional qualitative comparisons between the Stage I results and our final results. Clearly, Stage I of DDM$^2$ alone cannot restore indicative geometries and textures of the brain. With our Stage II and Stage III, the diffusion model is able to generate such missing contents for more promising denoising results.

**Noise residual comparisons.** In Figure 21 we present noise residual comparisons between Patch2Self and DDM$^2$. Our method suppresses more noise and does not show any anatomical structures in the corresponding residual plots.

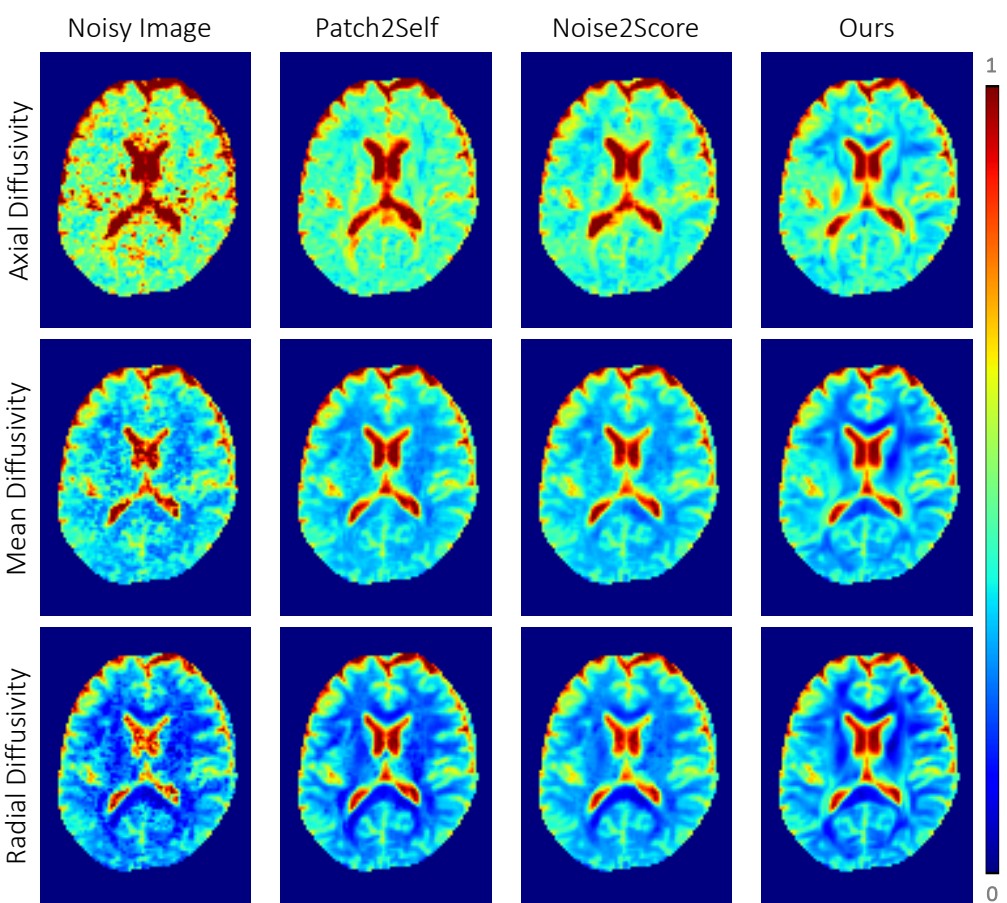

Figure 16: Comparison results of Axial Diffusivity, Mean Diffusivity, and Radial Diffusivity.

Figure 17: Additional qualitative results. DDM$^2$ demonstrates superior denoising quality in both image restorations and faithful reproduction of the original image contrasts. Arrows depict regions of hyperintense signal that may be caused by additional anatomical structure being represented by the other diffusion directions. Best viewed in zoomed in.

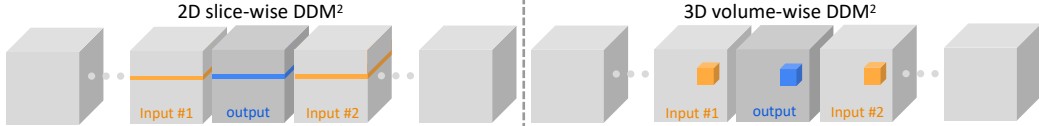

Figure 18: Intuitive comparisons between **Left:** 2D slice-wise DDM$^2$ (adopted) and **Right:** 3D volume-wise DDM$^2$.

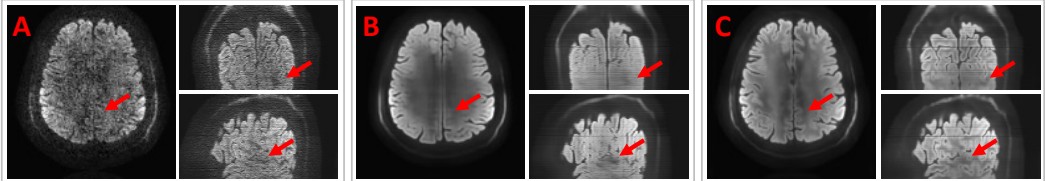

Figure 19: Additional comparisons at the transverse, Coronal, and Sagittal planes. **A:** Noisy input. **B:** results of volume-wise DDM$^2$. **C:** results of slice-wise DDM$^2$.

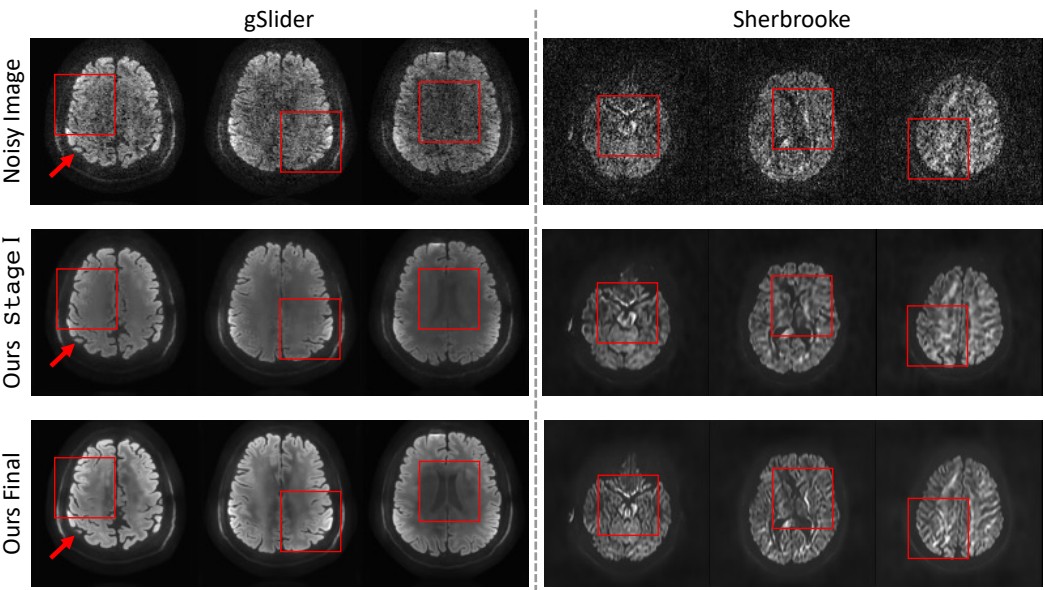

Figure 20: Additional comparisons between DDM$^2$ `Stage I` results and DDM$^2$ final results on our gSlider dataset and the Sherbrooke dataset. Arrows indicate possible artifacts caused during denoising, potentially due to partial volume effects affecting the edges of the grey matter or possible hallucinations induced by the denoising blurring. Major differences are highlighted.

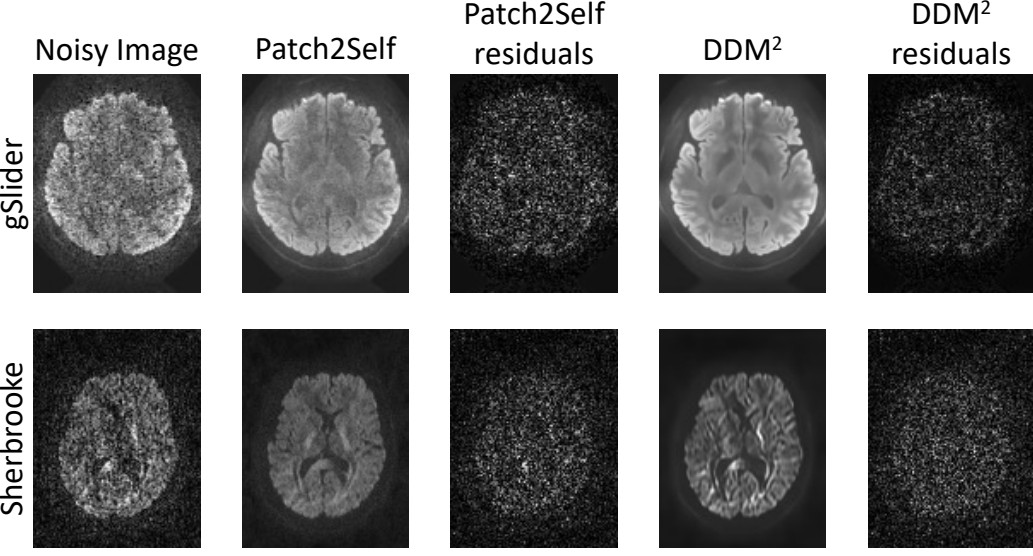

Figure 21: Residual noise comparisons between Patch2Self and DDM$^2$.

