# OpenReview forum: "DDM$^2$: Self-Supervised Diffusion MRI Denoising with Generative Diffusion Models"
_ICLR.cc/2023/Conference — ICLR 2023 poster_

### Official Review · Reviewer_RsQ4 · 2022-10-25

**Confidence:** 5
**Correctness:** 1
**Technical Novelty And Significance:** 2
**Empirical Novelty And Significance:** 2
**Recommendation:** 1

**Clarity, Quality, Novelty And Reproducibility:**


The work is original but the paper lacks clarity and quality of analyses. A key shortcoming is the validation is done only on the raw signal. The analyses needs to happen on the derivative microstructure measures or tractography tasks.


**Strength And Weaknesses:**

Strengths

The proposed method makes use of diffusion models to perform the denoising. While the approach seems useful, there are some questions which need to be answered:

Weakness

The official implementation of Patch2Self does not have MLP as a regression model. It is stated clearly in the GitHub repository of the Patch2Self implementation that DIPY implementation is the official one. Was this the one that was used? If yes, please can you provide the model parameters set?
In the supplement of the Patch2Self paper, the 1st figure shows that MLP performs worse than OLS and Ridge. Why was this chosen as the regression model for comparison?
The authors of the paper compare the performance of DDM2 with DIP, Noise2Noise and Noise2Score. DIP and Noise2Noise are not valid for dMRI data on may accounts. Was care taken to ensure that the training was performed on dMRI data? I would assume that these models would not do a good job out-of-the box.

The authors say that the Patch2Self was compared in all cases with patch radius 0 and MLP model. But as per the Patch2Self paper (Fig. 6C), if the number of volumes is lower, the patch radius needs to be increased. This makes the comparison in Fig. 5 incorrect.  The official implementation of Patch2Self needs at least 10 volumes for optimal performance.

It is also unclear why 2D slice-to-slice would not have structure in the residuals as opposed to 3D volume-to-volume mapping. This needs to be discussed in further detail.

In Figure 15, for the Sherbrooke dataset, the residuals are flipped -- please rectify this mistake.

The authors do not do a good job on validating the performance of the denoiser. While FA is an important metric the way SNR and CNR is computed is invalid. Furthermore the comparison needs to be done on different microstructure measures.

Generative models typically run the risk of hallucinating structures -- which may not be visible in the residuals. An analysis showing this needs to be added. This can be shown via DEC FA maps and Tractography analyses.


**Summary Of The Paper:**

The paper proposes a new method for denoising diffusion MRI data using recent advances in reverse diffusion. The method proposed a three stage approach and builds on top of Patch2Self which is the state-of-the-art method in the field of Diffusion MRI. The method is developed using generative models which indeed show potential for denoising applications.

**Summary Of The Review:**

While the method is new, it has quite a few shortcomings in its current form. Answering the above questions and improving the validation analyses should fix the issues.

---

> ### Author Response · Authors · 2022-11-13
> **Response to Reviewer RsQ4**
>
> **Our manuscript & supplementary have been revised. All major manuscript changes are shown in blue.**
>
> > **Patch2Self comparison**
>
> The official Patch2Self implementation doesn’t provide customized options for model architectures. For fair comparison to maintain the number of parameters between DDM$^2$ and Patch2Self, we reimplemented MLP-based Patch2Self in our paper. Here we present new experiments with all available denoisers offered in official Patch2Self implementations in **Appendix Sec. I, first paragraph**. There is no clear distinction between different Patch2Self denoisers in terms of denoising quality yet at the same time, we maintain parity in our comparisons. Overall, DDM$^2$ surpasses all official Patch2Self denoisers significantly.
>
> > **DIP/N2N Appropriateness**
>
> Deep Image Prior relies on the assumption that a convolutional neural network can fit to an underlying signal but not its noise, to implicitly perform denoising during an autoencoding task. This fits the formulation for denoising a diffusion MRI scan or any arbitrary image with noise present. Noise2Noise relies on having two independent observations fo the same signal but with different noise characteristics. Diffusion MRI with different diffusion directions fits this requirement. Both methods have been used previously for denoising of diffusion MRI data. DIP [a] and Noise2Noise [b]. We applied the same data preprocessing procedures to all of the comparison methods before training them from scratch.
>
> > **Patch2Self Radius/Performance**
>
> We revised our Fig. 5 and updated Patch2Self results obtained with radius=1 and Ridge model. We noticed that the improvement brought by larger patch radius is limited when the number of volumes is low. We also experimented with radius>1, and no obvious differences can be observed on denoising results but computations increased cubicly.
> Furthermore, even with increased patch radius, Patch2Self had worse performance with a fewer number of diffusion volumes compared to DDM$^2$. Since many clinical diffusion MRI scans use <10 diffusion directions, DDM$^2$ presents an additional improvement over this weakness of Patch2Self. If the optimal Patch2Self performance occurs in scenarios that are not clinically realistic, its utility is substantially lowered.
>
> > **2D vs 3D Residuals**
>
> This has been discussed in the “3D Volume-wise DDM$^2$” section. A residual image (i.e. a difference image) between a noisy input and its denoised version would be noise-like in the case of a perfect denoiser because the denoiser maintains all image structure and only removes noise. The more imperfect a denoiser, the higher the structure that remains in the residual image. We describe visually in Figure 6 and through text in “3D Volume-wise DDM$^2$” that the performance of the 3D denoiser (Figure 6B) is worse than the performance of the 2D denoiser (Figure 6C). Consequently, the 3D residual image maintains more structure (Figure 6D) because it is a worse denoiser than the 2D DDM$^2$ (Figure 6E). Furthermore, in the same section, we posit multiple hypotheses why this would be the case.
>
> > **DDM$^2$ Validation**
>
> SNR and CNR were calculated in an identical manner to the prior Patch2Self manuscript from NeurIPS 2021 where the mean signal from a region of interest is divided by the variance of a background noise-only section, as is done in prior MRI studies [c]. We have now added additional DTI measurements of Axial Diffusivity (AD), Mean Diffusivity (MD), and Radial Diffusivity (RD) in **Appendix Sec. J, first paragraph**. AD/MD/RD are quantitative metrics that are comparable across the methods while DEC FA maps are more challenging to compare in the absence of a higher-quality ground truth.
>
> > **Additional validations**
>
> Besides the validation on the actual diffusion MRI scans, we also perform a validation of the generated DTI metrics of fractional anisotropy, as well as axial, mean, and radial diffusivity. We perform SNR and CNR analyses of regions of interest. We further perform new validations on synthetic data created by adding realistic zero-mean complex Gaussian noise to actual k-space measurements (**Appendix Sec. H**).
>
> [a] Lin, Y. C., & Huang, H. M. (2020). Denoising of multi b-value diffusion-weighted MR images using deep image prior. Physics in Medicine & Biology, 65(10), 105003.
> [b] Kawamura, Motohide, et al. "Noise2Noise MRI for High-resolution Diffusion-weighted Imaging of the Brain: Deep Learning-based denoising without need for Highly Averaged Ground-truth Images."
> [c] Jarrahi, Behnaz, and Sean Mackey. "Characterizing the effects of mr image quality metrics on intrinsic connectivity brain networks: A multivariate approach." 2018 40th Annual International Conference of the IEEE Engineering in Medicine and Biology Society (EMBC). IEEE, 2018.

---

### Official Review · Reviewer_Bjyp · 2022-10-25

**Confidence:** 3
**Correctness:** 3
**Technical Novelty And Significance:** 3
**Empirical Novelty And Significance:** Not applicable
**Recommendation:** 6

**Clarity, Quality, Novelty And Reproducibility:**

- The manuscript is written quite clearly.
- Most of the components should be able to be reproduced.
- Differences from priors are well addressed.

**Strength And Weaknesses:**

Strength:
- Clear text and step-by-step description of the method.
- Differences from prior works well explained.

Question (rather than weakness):
- It would be nice to have an explanation how prior denoising of MIR are performed with supervision, as typical denoising methods for images are unsupervised.
- In Stage I, patches are replaced by image slices and slice-to-slice mappings are learned. There must be drawbacks from such a coarse mapping (e.g., losing local structures in the image space) and I am wondering if the authors have any thoughts on this.
- Is there any backbone model to approximate necessary functions within the framework?
- What are $\lambda$ in equation (4)?
- In the method, it is not clear where the self-supervised training is occuring.
- Is the noise in typical MRI or diffusion MRI iid or systematic? The noisy images in the experiments are synthesized with different $\beta$ levels (correct me if I misunderstood) and I am not sure if they truly mimic the noise pattern in reality.

**Summary Of The Paper:**

This paper proposes a self-supervised denoising method for diffusion MRI with diffusion model. The model consists of three steps which 1) learn noise pattern, 2) match an input to different diffusion stages and 3) diffusion model that generate denoised image. The model is validated on 4 different dataset to demonstrate generalization capability of the method.

**Summary Of The Review:**

I am not a super expert in diffusion model so I cannot judge very well on the novelty, but I think the differences of this work from the prior ones are well explained.
Text is written clearly to demonstrate the authors' ideas and they are well validated through the experiments.

---

> ### Author Response · Authors · 2022-11-13
> **Response to Reviewer Bjyp**
>
> **Our manuscript & supplementary have been revised. All major manuscript changes are shown in blue.**
>
> > **Explanation of prior supervised denoising**
>
> We have now added a description of prior supervised denoising approaches for diffusion MRI. Namely, in Section 2.1 (Diffusion MRI), we add the following: “These (supervised) methods scan the same volume multiple times, average these multiple low-SNR images to generate a high-SNR image, and subsequently, train a supervised model to transform a single low-SNR image to the averaged high-SNR image.”
>
> > **Patch- vs Image-Based Stage I**
>
> This is a good point. There is a fundamental tradeoff between gaining long-range dependencies and global context with image-wise training versus losing local fine-grained detail with a smaller patch-wise training. We decided to go image-wise training inspired by the underlying physics of the MRI acquisition process. Noise in MRI can vary spatially, however, the spatial frequencies that alter this SNR (driven by the coil sensitivity maps of the MRI receive coil used) vary slowly spatially. Limiting our Stage I training to small patches (like in a vision transformer) may highlight local structures better, but at the expense of non-local image pixels that also share similar SNR characteristics that can be beneficial for denoising. Overall, while our choice was driven by the underlying physics, we agree that a systematic ablation study may be beneficial here to carefully evaluate patch sizes to find the ‘sweet spot’ between local and global context. This study can start with very local features (i.e. 16x16 patches) and increase systematically to the whole image. Due to computational constraints, we can include this in the camera ready supplementary version.
>
> > **Backbones**
>
> We are unclear whether the reviewer asked about model architecture backbones or the or any backbones for the equations we describe? If it is the former, we adopt a commonly used U-Net arch as the model architecture backbone for Stage I and III of the DDM$^2$ denoising process. If it is the latter, we enforce the Stage II noise estimate to have a zero-mean Gaussian distribution as a mathematical backbone for our subsequent processes. If the reviewer could clarify this point, we can try to further address the specific issue.
>
> > **Equation 4 $\lambda$**
>
> $\lambda$s are scale coefficients, which have been first introduced right after equation (1). We added  \lambda into the table of symbols (Table 1) for improved clarity.
>
> > **Self-Supervised training location**
>
> We apologize for the confusion. We supervised the denoising of the volume $\mathbf{x}$ by inputting the noisy priors {$\mathbf{x}’$} (also known as supporting data) via J-Invariance optimization in Stage I and Stage III. Since there are no ground truths available and both $\mathbf{x}$ and {$\mathbf{x}’$} come from the same MRI acquisition, our whole framework is optimized in a self-supervised manner.
>
> > **Noise patterns in presented images**
>
> All datasets considered in the paper are real-world data that have been previously acquired from research or clinical studies on actual MRI scanners. These noisy acquisitions we experimented with, all directly represent real-world noise patterns in MRI. To address the reviewer question, noise in foreground tissues MRI has a systematic occurrence whose intensity is modulated by the spatial location in the image (with respect to the receive coil used). The \beta s are calculated for each image during Stage II, and they are not used for image synthesis. Furthermore, for completeness and according to reviewer Bjyp and M8jP suggestions, we do add a new experiment where we synthetically create noisy images from raw k-space MRI data, as described in Appendix Sec. H. In this experiment, we observe that DDM$^2$ maintains stability even for very noisy scans, while outperforming Patch2Self in all noise levels.

---

### Official Review · Reviewer_Wzbi · 2022-11-01

**Confidence:** 4
**Correctness:** 4
**Technical Novelty And Significance:** 3
**Empirical Novelty And Significance:** 3
**Recommendation:** 6

**Clarity, Quality, Novelty And Reproducibility:**

The approach is clear and appears to be novel. The quality is high and the work appears to be reproducible. I commend the authors for their effort in building a clear and transparent work.


**Strength And Weaknesses:**

Strengths:
- The landscape of self-supervised image denoising is quite crowded and can be difficult to navigate. The authors do a pretty good job contextualizing their work within this space.
- The approach to training a conditional diffusion model through spatially shuffling the noise residual is interesting.
- The ablation results lend credibility to the need for the three-stage approach, differentiating it from other self-supervised denoising works
- Extensive experiments are conducted on different datasets
- The paper is very detailed and code to reproduce results are provided

Weaknesses:
- It is unclear how this work fits in with other denoising methods such as [1] and [2]. For example, as the noise is assumed Gaussian, could [1] simply be used to denoise the volumes. Furthermore, could the denoiser in [1] be used according to the strategy in [2] to further refine the solution? To me this seems more straightforward than the proposed three-stage approach.
- How would the approach compare to running annealed Langevin dynamics using a pre-trained MRI diffusion model (for example, from [3]) with an identity A operator? In other words, is the scan-specific nature of the method necessary?
- As the approach is not actually specific to diffusion MRI, and the authors report that even n=1 is sufficient for learning the denoiser, could a different MRI dataset (where GT is known) be used to validate the methods (with noise added on top of the GT)? For example, from FastMRI or SKM-TEA?
- It is unclear from reading the paper if multiple diffusion directions is necessary
- It seems like all the datasets have multiple scans at the same b-value (for example 10 have b=0 and 60 have b=1000). Are these b-value images collected with different diffusion directions? What would happen if the images with the same b-value and direction are simply averaged together? Could this be used as a GT, either for comparison or for some other supervised approach?
- It is unclear how the dimensions of the data correspond to the directions/b-values. Please explain the dimension sizes as they relate to the # of directions and b-values.
- is the method only applicable to denoising with multiple b-values? do the b-values need to be the same for denoising to work?
- It is said that two neuroradiologists reviewed the images. Can you please comment more on this evaluation? How many images were reviewed per dataset? why wasn't a larger evaluation conducted using the neuroradiologists, for example to rate image quality, SNR, CNR, fine details, etc?
- Could the authors test (or at least comment) on the impact of Rician vs. Gaussianity, for instance with GT data and additive noise? It was unclear in the text whether the data used are complex-valued or magnitude only, and what the impact is of the Gaussianity assumption.

[1] Christopher A Metzler, Ali Mousavi, Reinhard Heckel, Richard G Baraniuk, Unsupervised learning with Stein's unbiased risk estimator, arXiv preprint arXiv:1805.10531
[2] Zahra Kadkhodaie, Eero P. Simoncelli, Solving Linear Inverse Problems Using the Prior Implicit in a Denoiser, https://arxiv.org/abs/2007.13640
[3] Ajil Jalal, Marius Arvinte, Giannis Daras, Eric Price, Alexandros G Dimakis, Jon Tamir, Robust compressed sensing mri with deep generative priors.  Advances in Neural Information Processing Systems


**Summary Of The Paper:**

This paper provides an approach for self-supervised denoising of MRI data using diffusion-based generative models. The approach is applied to denoise diffusion-weighted MRI, hence the name DDM2. A three-stage approach is used to train/denoise: 1. learning a denoising function with existing self-supervised approaches; 2. fitting the noise statistics to a Gaussian; and 3. training a conditional diffusion model using the noisy data and noise statistics. The method is evaluated on four different datasets of diffusion-weighted MRI. Comparisons to other self-supervised methods are shown, as well as ablation experiments.


**Summary Of The Review:**

The authors develop a three-stage approach to denoising diffusion-weighted MRI using self-supervised denoising diffusion models. The method appears to improve over other self-supervised baselines, though some details remain unclear. The contribution is clear and the work is well-organized and novel.

---

> ### Author Response · Authors · 2022-11-13
> **Response to Reviewer Wzbi (1/2)**
>
> **Our manuscript & supplementary have been revised. All major changes are shown in blue.**
>
> *Due to the length limit per comment, for our responses on simulation experiments and impact of Rician vs Gaussianity, please see the followup comment section or our responses to reviewer M8jP.*
>
> > **Comparison with [1,2]**
>
> Although Gaussian noise are assumed, we found that DDM$^2$ can also learn data-specific noise distributions. We updated comparison results in the **Appendix Sec. I, 2nd paragraph**. DDM$^2$ shows clear improvements over the other methods.
>
> > **Necessity of scan-specific method**
>
> This is a good point. Diffusion models generate data based on learnt prior information. When training on mixed data sources, diffusion models may duplicate anatomical patterns from one scan to recover another and potentially induce hallucinations. Scan-specific methods in this regard, are unbiased from a data driven perspective, and will only generate details that can be inferred from this particular scan.
>
> > **Multiple Directions Requirement**
>
> It is not a strict requirement to have multiple directions. In the case of DTI, the multiple diffusion directions indicate near identical image signal and noise statistics. For DDM$^2$ and Patch2Self, we simply need multiple observations of the same signal under different noise conditions, which the notion of multiple diffusion directions suffices. However, if multiple averages were available (see *Denoising across b-values* below), that would also suffice.
>
> > **Averaging B-values**
>
> It is correct that the different datasets have the same b-values for different diffusion directions. Since we do not use the b=0 scans in this work, we now actually exclude reporting those to improve clarity. It is also entirely correct that if multiple averages for the same diffusion direction and b-value were available as separate imaging volumes, it could serve as a nice ground truth. This would allow evaluating denoising of a single image, compared against the combined images. However, no public datasets have such data separated per image average since MRI scanners commonly combine the multiple averages into one image series during the routine reconstruction. As a surrogate evaluation, our results on simulated data can be used to demonstrate the superiority of our method when GTs are available.
>
>
> > **B-values and Directions**
>
> We have now made the Datasets section more clear by reporting all datasets with the following nomenclature: (X x Y x Z x Diffusion Directions) and specifying the corresponding b-values. To clarify, gSlider, Sherbrooke, HARDI, and PPMI datasets had 50, 193, 150, and 64 diffusion images with b-value>0, respectively. The b-values used for denoising in those datasets are 1000, 1000, 2000, and 2000, respectively.
>
> > **Denoising across b-values**
>
> Thank you for raising this interesting question. We initially focused on homogeneous b-values since these represent the most clinical and research MRI protocols (where one b-value shell is sampled across multiple directions). However, newer q-space MRI sequences and even the Sherbrooke dataset that we used indeed samples multiple b-values. During this rebuttal duration, we tried including b=1,000 images to help guide b=2,000 denoising (**Appendix Sec. I, 3rd paragraph**). Our preliminary results show that the resultant images, although denoised, have a high amount of blurring. This is most likely due to the fact that different b-values have different gradient strengths, and as a result, different distortion artifacts induced by the echo planar readout. This distortion across b-values manifests as blurring in the denoised image. Thus, based on the experiments that the reviewer suggested, we believe that including different b-values is an interesting research direction, however, additional work beyond this week-long rebuttal duration will be necessary to account for distortions between b-values.
>
> > **Neuroradiologist Review**
>
> This is a good point and something we had requested from our neuroradiologists initially. However, their response was that diffusion MRI scans are seldom evaluated in isolation without a paired high-resolution anatomical sequence (T1-weighted MRI, for example). This is especially the case since diffusion MRI may contain eddy-current and echo-planar-imaging-based distortions for accurately assessing fine anatomical details. To enhance reproducibility of our work, we intentionally worked with publicly-available DTI datasets, however, these were not paired with anatomical sequences. Consequently, we chose to delay a full reader study for an ongoing clinically-focused follow-up study where we sample both multi-average DTI data as well high-resolution T1, T2, and FLAIR sequences.

---

> ### Author Response · Authors · 2022-11-13
> **Response to Reviewer Wzbi (2/2)**
>
> > **Evaluation on simulated data**
>
> Thank you for your suggestions. We have done exactly that using the SKM-TEA knee MRI dataset that has raw k-space data available. With SKM-TEA, we add additional complex zero-mean Gaussian noise to the k-space per coil, followed by the routine reconstruction pipeline. These noisy acquisitions perfectly simulate acquiring multiple noisy averages of the same anatomy, as was suggested by the reviewer. Experimental details and results are updated in **Appendix Sec. H**. Specifically, we generate acquisitions images with varying SNR levels. We find that in a low-noise setting, DDM$^2$ and Patch2Self demonstrate similar performance, however, as noise levels increase (decreased SNR), Patch2Self cannot denoise the images, while DDM$^2$ still depicts high-fidelity denoising (Fig. 11 & 12).
>
> Through these two figures we further show that DDM$^2$ performance when using n=1 and n=10 supporting volumes is near identical, whereas Patch2Self requires n=10 to have better denoising performance (that is still inferior to ours). This observation highlights the data efficiency of our proposed approach since it could be applied to datasets with very few supporting volumes, as low as 1, which are very common clinically.
>
> Overall, these synthetic experiments successfully depict robustness of DDM$^2$ to varying noise levels and data efficiency. We thank the reviewer for pushing us to perform these.
>
> > **Gaussian vs Rician Noise**
>
> The newly added simulated experiments in Appendix Sec. H substantially addresses this question. All complex noise in MRI is Gaussian, however, the manifestation of complex Gaussian noise in the background regions of magnitude images is Rician. Thus, instead of simulating Rician noise on magnitude images that we initially worked with, we simulated complex noise on SKM-TEA raw k-space data, resulting in realistic noise distributions in the reconstructed magnitude images. Figures 11 and 12 depict adequate denoising in the foreground and background regions that exhibit Gaussian and Rician noise, respectively. This depicts that the Stage I and III denoising functions may be adaptable to different noise patterns.

---

### Official Review · Reviewer_M8jP · 2022-11-01

**Confidence:** 4
**Correctness:** 4
**Technical Novelty And Significance:** 3
**Empirical Novelty And Significance:** 4
**Recommendation:** 8

**Clarity, Quality, Novelty And Reproducibility:**

The article presents novel content and is clear and the authors have made an effort to summarize the state-of-the-art and comparing techniques very well. I appreciated that. Although I have not found the link to the code, I believe that it will be sent out at publication. Link to the data is present, and tips on how to make everything to work are also present. Therefore, I believe that the article is reproducible.

Minor clarifications:
0) in the introduction of diffusion MRI, the gradients are not magnetic fields, but rather generate a spatially-varying magnetic field.
a) Page 2 → and uses them TO supervise each other
b) figure 1 → and black blocks indicate.. you mean blue blocks?
c) Page 4 → however 3D volume results are shown in the supplementary MATERIAL
d) I don’t understand the right-hand-side of eq.4 Where does it come from? Please give indications on its derivation.
e) The sentence at page 5 “We can determine the closest matching state S t of x by comparing G and posterior p(S t )—σ and β t .” is not intelligible. Please rephrase.
f) Page 7 → please fix the sentence “We do not that N2S did have improved worst-case results than DDM 2 but this was likely”
g) Page 7 → please fix sentence “are necessary for DDM 2 Ṫhis by its nature”

**Strength And Weaknesses:**

The article is interesting, inspiring and introduces a series of improvements that are of interest for the community, as also is the shown performance. I like the formulation of the solution to the problem in three stages and the advantage of each of them is clearly presented with quantitative information.

The main weakness remains, in my opinion, the lack of synthetic validation. Despite many "real data" examples it is not possible to assess the amount of allucinations generated by the methodology and the amount of intensity bias introduced with each pixel.

In my experience with Patch2Self on synthetically generated data I could prove that the estimation of quantitative measurements (e.g. the estimation of diffusivities) is subject to a bias, which is not present, for instance, with methdos based on MPPCA. In this sense, without synthetic results it is not possible to assess this kind of performance.

I note, especially in figure 11 and 14 , that indeed the DDM2 method may introduce relevant allucinations. For instance, in fig. 11 please look at the first raw, and observe the cerebrospinal fluid areas in the middle of the brain, which have very different shapes compared to the noisy image. This is an example of "invented" tissue contrast. Another example is found by looking at the first column image of each row of figure 14 (left hand side). In the bottom left part of the brain (just beneath the red square) there is an "island" of tissue contrast that is better preserved in the stage 1 compared to the final result.

I think that the authors should point out with arrows to this kind of differences. While reading I had the impression that authors were stressing a bit too much the fact that there were very few allucinations, and in my opinion, that is not the case. In fact, I strengthen the comment about the lack of synthetic experiments!


I have other points that I would like the author to address:
- Does the presented method generalize to the case when the noise variance varies spatially? In my opinion it does not, so please mention that a limitation.
- Noise in MRI is spatially correlated (use of Partial Fourier, for example): what are the consequences of this for the approach?
- Noise in MRI is non-Gaussian (e.g. Rician): how does this affect the proposed method which clearly only works for Gaussian data? In section 5 authors mention SENSE, which does not perturb the Gaussian nature of the noise on complex images, but still yields Rician magnitude images!


**Summary Of The Paper:**

The paper proposes a new method for the denoising of diffusion weighted images with diffusion denoising models. It brings about a series of modifications on existing literature that actually determines higher performance compared to state-of-the-art.

The article presents extensive validation which is highly appreciated, and shows results from comparing state-of-the-art techniques. My main concern with this is the lack of synthetic data validation, which is something that (unfortunately) many articles on the subject lack. I think that when talking about denoising it would be somehow mandatory to display synthetic results, even more when denoising is achieved through machine learning, where the risk of introducing "artefactual allucinations" in the images is very high.



**Summary Of The Review:**

The article tackles a challenging problem. It comes up with new ideas/modifications with respect to the state of the art. It achieves remarkable results although the full extent of these is difficult to evaluate. This is not just a problem of this article but of all the comparing state-of-the-art works, since they do not provide results on synthetic data. This is a problem with machine learning/generative techniques for denoising of medical data, because allucinations can cause dramatic effects. I would not use this denoising method on data from patients. Nevertheless, the methodology described in the article is certainly of interest for the community and can inspire other researchers. As such, the article has a great value and represent a step forward in the good direction. Despite synthetic validation is missing, favorable comparison with many competing methods has been clearly shown.

To summarize, the article discloses some key ideas to solve the denoising problem using diffusion methods. These ideas are valuable and of interest, and lead to improved performance. The problem statement is clearly formulated, and the literature well acknowledged. I recommend  the acceptance of the article as is. If a second revision stage is foreseen, I would recommend the authors to take into account my comments.

---

> ### Author Response · Authors · 2022-11-13
> **Response to Reviewer M8jP**
>
> **Our manuscript & supplementary have been revised. All major manuscript changes are shown in blue.**
>
> > **Evaluation on simulated data**
>
> Thanks for the suggestion, which was also echoed by other reviewers. We use the SKM-TEA knee MRI dataset that has fully-sampled k-space data available and we add additional complex zero-mean Gaussian noise to the k-space per coil, followed by the routine reconstruction pipeline. These noisy acquisitions perfectly simulate acquiring multiple noisy averages of the same anatomy, as was suggested by the reviewer. Experimental details and results are updated in our **Appendix Sec. H**. Specifically, we generate acquisitions images with varying SNR levels. We find that in a low-noise setting, DDM$^2$ and Patch2Self demonstrate similar performance, however, as noise levels increase (decreased SNR), Patch2Self cannot denoise the images, while DDM$^2$ still depicts high-fidelity denoising (Fig. 11 & 12). Overall, these synthetic experiments successfully depict robustness of DDM$^2$ to varying noise levels. We thank the reviewer for pushing us to conduct these experiments which depict the robustness and data-efficiency of DDM$^2$.
>
> > **Possible Hallucinations**
>
> This is a fair point. We have updated the figures as suggested, added arrows to point out these possible artifacts, and re-worded our claims regarding lack of possible hallucinations. In our manuscript, we posit that such variations may be induced either by the model itself in trying to reconstruct anatomical information using other diffusion direction information or due to partial volume artifacts at the edges of the grey matter. In the future, we will investigate whether only including b-vectors closest to the b-vector being denoised may limit such artifacts. We agree that a follow-up systematic review of such images will be beneficial. These observations further emphasize the synthetic experiments that we newly performed so that we can quantitatively and qualitatively assess the fidelity of the denoising.
>
> > **Spatially Varying Noise**
>
> Image SNR and the underlying signal/noise characteristics vary as a function of the underlying coil sensitivities of the MRI receiving apparatus. In our synthetic experiments, we depict how DDM$^2$ performs denoising consistently across a majority of the anatomy based on the corresponding residual maps. The variation in denoising is likely driven by the underlying tissue SNR, rather than the exact location of the voxel in a given image. The consequence of spatially varying noise is that it may be necessary to craft DDM$^2$ to include a large receptive field to account for noise that varies based on the spatial location. Consequently, this discourages the use of 3D patch-based methods since they may not be able to learn the general signal and noise structure across scales. A similar argument may be made regarding vision transformers-based approaches that lack global image context.
>
> > **Guassian vs Rician Noise**
>
> Great point that can be explained thanks for your comment regarding the need for additional synthetic experiments! Rician noise in magnitude images is a consequence of low-intensity zero-mean complex-value Gaussian noise benign transformed into magnitude images. As is visible in these Fig. 11 and 12 for these synthetic experiments, we see comparable denoising performance in the foreground and background regions (where we expect Gaussian and Rician noise, respectively). However, additional studies that require more duration than the rebuttal period could further evaluate these hypotheses quantitatively.
>
> > **Minor clarifications**
>
> Thanks for your corrections, we updated them in our paper.

---

### Decision · Program_Chairs · 2023-01-20

**Decision:**

Accept: poster

**Justification For Why Not Higher Score:**

The reviewer scores do not support it.

**Justification For Why Not Lower Score:**

This should actually be a relatively clear accept; the one rejection recommendation clearly has to do with the unfair comparison with state of the art. The authors have updated this in the rebuttal phase, and as long as they update the paper accordingly, the paper can be accepted.

**Metareview: Summary, Strengths And Weaknesses:**

Summary:
This paper suggests a new method for denoising diffusion MRI images based on diffusion models. As can be seen from the reviews, one of the main objections to the paper, by Reviewer RsQ4, is that the comparison with existing baselines was not carried out in a fair way. In their rebuttal, the authors have carried out further experiments, and after a discussion with reviewer M8JP, they have suggested a more sober description of their results. It is a requirement for the acceptance of the paper that this description is also added to the paper.


Strengths:
- The paper is interesting, well written and novel
- The experimental section is strong, given the updates in the rebuttal


Weaknesses:
- As explained above, in the original version of the paper, the comparison with existing work was not fairly carried out; however, the authors rectified this in their rebuttal
- Most of the weaknesses listed by the reviewers are actually questions, which I take to be more of a sign of interest than a weakness.

**Note From Pc:**

if the above contains the word "oral" or "spotlight" please see: "oral" presentation means -> notable-top-5% and "spotlight" means -> notable-top-25%. As stated in our emails, we are disassociating presentation type from AC recommendations